



# Development of a subglacial lake monitored with radio-echo sounding: Case study from the Eastern Skaftá Cauldron in the Vatnajökull ice cap, Iceland

Eyjólfur Magnússon[1], Finnur Pálsson[1], Magnús T. Gudmundsson[1], Thórdís Högnadóttir[1], Cristian Rossi[2],
Thorsteinn Thorsteinsson[3], Benedikt G. Ófeigsson[3], Erik Sturkell[4], Tómas Jóhannesson[3]

[1]Institute of Earth Sciences, University of Iceland, Reykjavík, IS-102, Iceland
[2]Remote Sensing Technology Institute, German Aerospace Center (DLR), Wessling, 82234, Germany
[3]Icelandic Meteorological Office, Reykjavík, IS-105, Iceland
[4]Department of Earth Sciences, University of Gothenburg, Gothenburg, Box 460, 405 30, Sweden

*Correspondence to*: Eyjólfur Magnússon (eyjolfm@hi.is)

**Abstract.** We present repeated radio-echo sounding (RES, 5 MHz) on a profile grid over the Eastern Skaftá Cauldron (ESC) in Vatnajökull ice cap, Iceland. The ESC is ~3 km wide and 50–150 m deep ice cauldron created and maintained by subglacial geothermal activity of ~1 GW. Beneath the cauldron and 200–400 m thick ice, water accumulates in a lake and is released semi-regularly in jökulhlaups. The RES record consists of annual surveys with 200–400 m between profiles in early summers of 2014–2020. Comparison of the RES surveys (2D migrated profiles) reveals variable lake area (0.5–4.1 km$^2$) and enables traced reflections from the lake roof to be distinguished from bedrock reflections. This allows construction of a digital elevation model (DEM) of the bedrock in the area, further constrained by two borehole measurements at the cauldron centre. It also allows creation of lake thickness maps and an estimate of lake volume at the time of each survey, which we compare with lowering patterns and released water volumes obtained from surface DEMs obtained before and after jökulhlaups. The estimated lake volume is 250x10$^6$ m$^3$ in June 2015 but 320±20x10$^6$ m$^3$ drained from the cauldron in October 2015. In June 2018, RES profiles reveal a lake volume of 185x10$^6$ m$^3$ while 220±30x10$^6$ m$^3$ was released in a jökulhlaup in August 2018. Considering the water accumulation over the periods between RES surveys and jökulhlaups, this indicates 10–20% uncertainty in the RES-derived volumes at times when significant jökulhlaups may be expected.

## 1 Introduction

Subglacial lakes are known in both temperate and cold glaciers. Floods originating from subglacial lakes, commonly referred to as jökulhlaups are well known and of variable magnitude. In warm bedded glaciers jökulhlaups are known to cause widespread and a manifold increase in basal sliding over periods of days (e.g. Einarsson et al., 2016), while a persistent leakage from such a lake caused significant deceleration of a glacier over a period of years (Magnússon et al., 2010). In Antarctica, water originating from subglacial lakes has been identified as a key cause of persistent fast-flow features (Bell et al., 2007; Fricker et al., 2007; Langley et al., 2011) as well as the cause of transient acceleration (Stearns et al., 2008).



Many subglacial lakes have been identified based on radio-echo sounding (RES) data. The first such observation was made more than 50 years ago (Robin et al., 1970), when RES data, acquired near the centre of East Antarctica, revealed a ~10 km long unusually flat subglacial surface with high reflectivity attributed to a "thick layer of water beneath the ice". Since then

RES has been used to identify hundreds of subglacial lakes, most of them beneath the Antarctic ice sheet. SAR interferometry and repeated altimeter surveys have in the past two decades revealed more than hundred areas of elevation changes within Antarctica that are associated with draining or filling of subglacial lakes (e.g. Gray et al., 2005; Smith et al., 2009), showing the dynamic nature of subglacial water systems. Many of these active lakes are, however, hard to detect from RES data and do not reveal a flat subglacial surface nor clear changes in reflectivity at the glacier bed (Carter et al., 2007; Siegert et al., 2014).

In some areas and particularly in Iceland, jökulhlaups originating from subglacial lakes are real natural hazards. Since the settlement of Iceland they have threatened the lives of people and livestock and repeatedly ruined farms, fields and vegetated areas. In the past century roads and bridges have been destroyed. This may explain why the recognition and research of subglacial lakes beneath Icelandic ice caps dates farther back than elsewhere (Thorarinsson and Sigurðsson, 1947; Thorarinsson, 1957). The subglacial lakes beneath the Icelandic ice caps owe their existence to localized geothermal activity

beneath the glaciers. The basal melting produces a depression in the glacier surface often referred to as ice cauldrons. This causes a low in the hydrostatic potential, which facilitates subglacial accumulation of meltwater both from the glacier surface and bed (Björnsson, 1988).

The three largest subglacial lakes in Iceland are located beneath the western part of the Vatnajökull ice cap (Fig. 1); Grímsvötn and the lakes beneath the two Skaftá cauldrons (denoted as Eastern and Western cauldron). Grímsvötn has been known for

centuries as a lake within Vatnajökull and the source of large jökulhlaups draining from Skeiðarárjökull outlet glacier in S-Vatnajökull although the exact location was not well known until identified in an expedition in 1919 (Wadell, 1920). Accounts describing jökulhlaups in the river Skaftá, probably draining from the Skaftá cauldrons, date at back to the first half of the 20$^{th}$ century (Björnsson, 1976; Guðmundsson et al., 2018). The first direct observation of the Eastern Skaftá cauldron (ESC) as a depression in the glacier surface is a photograph taken from an airplane in 1938. Aerial photographs taken by the U.S. Army

Map Service in 1945 and 1946 indicate that the Western Skaftá cauldron (WSC) did not exist at that time while ESC was much smaller than at present. The first known photographs showing WSC were taken in 1960 (Guðmundsson et al., 2018). The amount of water draining from these areas in jökulhlaups has, combined with information on surface mass balance, been used to estimate the power of the geothermal areas beneath Grímsvötn and the Skaftá cauldrons (Björnsson, 1988; Björnsson and Guðmundsson, 1993; Guðmundsson et al., 2018; Reynolds et al., 2018; Jóhannesson et al., 2020). These estimates result in a

geothermal power of 1500–2000 MW for Grímsvötn and similar power for the two Skaftá cauldrons combined, making these some of the most powerful geothermal areas in Iceland. Large scale melting by volcanic eruptions caused the most recent major jökulhlaups draining from Grímsvötn in 1938 and 1996, resulting in release of respectively 4.7 km$^3$ and 3.4 km$^3$ (Gudmundsson et al., 1995; Björnsson, 2002). In comparison the largest jökulhlaups from the Skaftá cauldrons are an order of magnitude smaller (Zóphóníasson, 2002; Egilsson et al., 2018).






**Figure 1 a:** The western part of Vatnajökull ice cap situated within the volcanic zones of Iceland (grey areas on inlet) and the locations of the Grímsvötn subglacial lake and the lakes beneath the Skaftá cauldrons. Jökulhlaups from the Skaftá cauldrons drain to the river Skaftá. Jökulhlaups from Grímsvötn drained until 2009 into the river Skeiðará (approximate position around the year 2000) and since then into the river Gígjukvísl. **b:** TanDEM-X DEM of the Eastern Skaftá cauldron (ESC) obtained a week after the jökulhlaup in 2015 represented as shaded relief (DEM location shown with red square in a). **c:** Sentinel 2 optical image of the same area as in **b** showing ESC almost ~3 months after the jökulhlaup in 2018 of. **d–e:** Photographs taken about 1 week after the 2015 (**d** by Benedikt Ófeigsson) and 2018 (**e** by Magnús T. Guðmundsson) jökulhlaups. The viewing angles are indicated with dashed red lines in **b** and **c.**



The setting at Grímsvötn is unique for subglacial lakes in Iceland, being located inside a triple composite caldera forming the
centre of the highly active Grímsvötn central volcano (Guðmundsson et al., 2013a). Most of the ice melting, volcanic and
geothermal, takes place near the caldera rims while the main water volume is stored near the centre of the main caldera. In
June 1987, low water level in Grímsvötn nine months after a jökulhlaup made it possible to map of the lake bed with RES and
additional seismic observations, possible (Björnsson, 1988; Gudmundsson, 1989). This, along with knowledge about the
thickness of the lake's glacier cover, allowed monitoring of the lake volume by measuring the surface elevation of the lake's
glacier cover near its centre (Björnsson, 1988; Gudmundsson et al., 1995). Such straightforward estimates leading to
reasonably accurate results are generally not possible for lakes beneath ice cauldrons, including the Skaftá cauldrons. The
interaction between intense melting at the bed and strongly converging ice flow, particularly when a cauldron is steep and deep
shortly after jökulhlaups, leads to great temporal and spatial variations in the glacier thickness above the lake. Hence there is
not be a clear relationship between the surface elevation within an ice cauldron and the volume of the lake beneath.

Monitoring the surface elevation of an ice cauldron provides useful information on the cauldron's behaviour, despite the lack
of quantitative results regarding the lake's volume. This particularly applies to studies of jökulhlaups causing significant
deepening of the cauldrons as mapping of the surface lowering can be used to measure the volume of the released flood water.
Regular measurements of the surface elevation near the centres of the Skaftá cauldrons have been carried out since the late
1990's (Guðmundsson et al., 2018; http://jardvis.hi.is/skaftarkatlar_yfirbord_og_vatnsstada). The elevation has been measured
with ground based GNSS measurements, repeated radar altimetry profiling from an airplane (Gudmundsson et al., 2007) and
from sporadic continuous digital elevation models (DEMs) obtained from various remote sensing data.

In Iceland, the attempts to survey water accumulation below ice cauldrons using changes in the elevation of reflective
subglacial surfaces from low frequency (5 MHz) RES data, were motivated by a swift, unexpected jökulhlaup from the
cauldrons of Mýrdalsjökull ice cap, S-Iceland, in July 2011 (Galeczka et al., 2014). It destroyed the bridge over the river
Múlakvísl, cutting the road connection along the south coast of Iceland for more than a week. The same RES profiles across
the Mýrdalsjökull cauldrons have been repeated, following the same path as accurately as possible, once or twice a year since
May 2012 with the aim of detecting if unusual water accumulation takes place beneath the cauldrons (Magnússon et al., 2017;
in review). This same RES survey approach has been carried out annually over ESC since June 2014. At that time jökulhlaups
had not been released from ESC for 4 years while the typical interval between jökulhlaups is 2–3 years (Guðmundsson et al.,
2018). The unusually long pause as well as the insignificant rise in cauldron surface elevation since summer 2011 were curious,
motivating the RES survey at ESC. In this paper the results of the RES study on ESC are presented. This includes a DEM of
the glacier (and lake) bedrock beneath the cauldron as well as an annual estimate the area, volume and shape of the lake in
2014–2020. We also present a unique comparison of the subglacial lake volume and shape in spring 2015 and 2018 with
elevation changes within the cauldron during two unusually large and destructive jökulhlaups, in autumn 2015 and in summer
2018 with maximum discharge of ~3000 $m^3$ $s^{-1}$ and ~2000 $m^3$ $s^{-1}$, respectively (Jónsson et al., 2018 and unpublished data of
the Icelandic Meteorological Office (IMO)). This gives a new insight into how rapid emptying (days) of a subglacial lake, with
fairly well known geometry, is reflected as elevation changes at the surface of the 200–400 m thick ice cover. The good



agreement between the lake volumes obtained from the RES surveys and from the surface lowering during the consequent jökulhlaup, as demonstrated below, shows the applicability of RES as a tool to monitor water accumulation in the lake, and
the potential hazard of jökulhlaups from the Eastern Skaftá Cauldron.

## 2 Data and Methods

### 2.1 Radar data

The RES data were obtained in early June or late May each year from 2014 to 2020 during the annual field trips of the Iceland Glaciological Society on Vatnajökull. The original profile grid over ESC first measured in 2014 consists of two sets of parallel
profiles (400–500 m between profiles), perpendicular to each other (Fig. 2a). This profile grid has since then been re-measured as accurately as possible every year (Fig. 2–4). Some parts of the profiles could not be measured in 2016 due to large crevasses formed during the 2015 jökulhlaup (Fig. 1b). Other parts of the survey profiles in 2017 and 2018 were defect due to artefacts in the survey caused by a supraglacial lake formed within the cauldron in the summer of 2016 and covered with snow the following winter (Fig. 5). Between the 2019 and the 2020 surveys, an englacial water body was probably formed tens of meters
below the glacier surface, affecting a substantial part of the 2020 RES data with similar artefacts as in 2017 and 2018 (Fig. 2h). The surroundings of the Skaftá cauldrons were specifically measured in 2017 and 2019. The density of the profile grid within the cauldrons was doubled (200–250 m between profiles) in 2018 (Fig. 4).

The RES data were acquired, following similar practice as for most RES surveys of Icelandic glaciers (e.g. Björnsson and Pálsson, 2020; Magnússon et al., in review), by towing with a snowmobile a low frequency transmitter (5 MHz centre
frequency) and a receiver unit on separate sledges, 35–45 m apart, with corresponding antennae in a single line. The snowmobile was equipped with a Differential Global Navigation Satellite System (DGNSS) receiver. The receivers used in this study were developed by Blue System Integration Ltd. (see Mingo and Flowers, 2010). The raw RES data are backscatter images where the x-axis corresponds to the number of the RES survey (256 or 512 stacked measurements). The y-axis is the travel time of received backscattered transmission relative to the triggering time of the measurement but receiver measurement
is triggered by the direct wave propagating along the surface from the transmitter. The centre position, $\mathbf{M}$, between transmitter and receiver for each RES survey was derived from the GNSS timestamp obtained by the receiver unit for each RES sounding, and the corresponding position of the DGNSS on the snowmobile projected back along the DGNSS profile by a distance corresponding to the half the antenna separation plus the distance from the RES receiver sledge to the snowmobile (~20 m). When surveying profiles without taking sharp turns the horizontal accuracy of $\mathbf{M}$ is expected to be <3 m but errors are mainly
due to variation in distance to the snowmobile, inexact timing of each RES survey (the sounding plus processing time of the stacked measurements is slightly varying but typically ~1 s) and the towed sledges not always accurately following the path of the snowmobile. The vertical accuracy is <0.5 m.







**Figure 2 a:** The initial RES survey route of ESC (location on corner inlet) in 2014. The DEM presented with shaded relief and contour map (20 m interval) was obtained from TanDEM-X data acquired 23 September 2015. **b–h:** An example of 2D migrated RES profiles for part of this route (from A to B on **a**) for all survey years. The vertical exaggeration is 2-fold. On each profile, the traced bed reflection (both from ice/bedrock and ice/water interface) and surface elevation are shown along with same information from the survey in the preceding year.

The strong direct wave is estimated as the average wave form measured with the RES over several km long segments and subtracted from the corresponding segment of raw RES measurement. The remaining backscatter, mostly from englacial and subglacial reflectors, is amplified as a function of travel time in order to have the backscatter strength as independent as possible of the depth to the reflectors. This along with the 3D location, **M**, for each measurement and corresponding transmitter and receiver 3D positions (half antenna separation, behind and in front of **M**, respectively, along the DGNSS profile) were used as inputs into 2D Kirchhoff migration (e.g. Schneider, 1978), programmed in Matlab (®Mathworks). The migration was




carried out assuming propagation velocity of the radar signal through the glacier, $c_{gl}$=1.68x10$^8$ m s$^{-1}$ (corresponding to $c_{gl}$ for

dry ice with density of 920 kg m$^{-3}$ (e.g. Robin et al., 1969); the choice of $c_{gl}$ and validation from borehole survey is discussed

in section 4.1.3) and 500 m width of the radar beam illuminating the glacier bed. This results in profile images as shown in

Fig. 2. The x- and y-axis of these images correspond to driven profile length and elevation in m a.s.l., respectively. The image

pixel dimension is dx=5 m and dy=1 m, corresponding roughly to the horizontal sampling density when measuring with ~1 s

interval at ~20 km hour$^{-1}$, and the 80 MHz vertical sampling rate (in 2014–2017; it is 120 MHz for a new receiver unit used in

2018–2020).

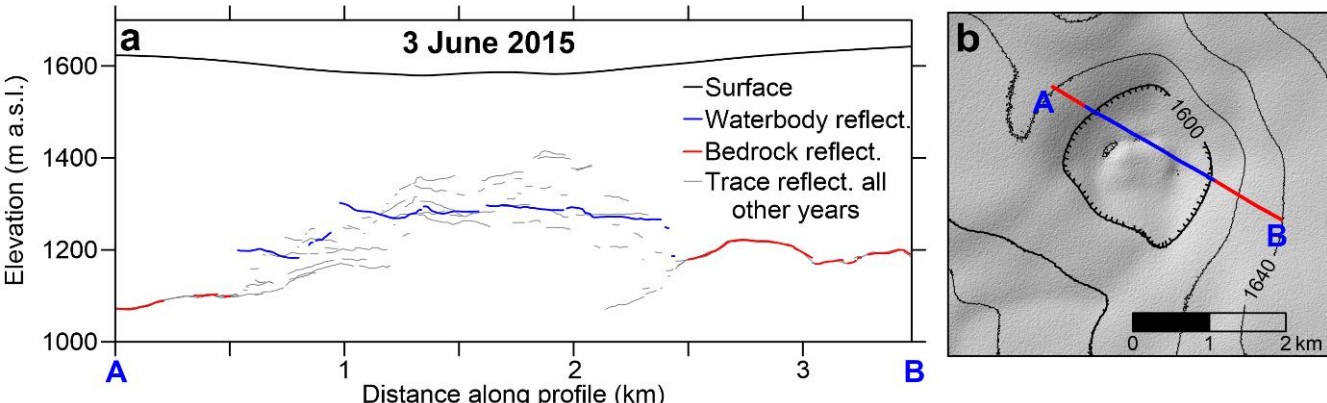

**Figure 3 a:** The traced reflections in 2015 (blue and red) for the same section of the RES survey route as in Fig. 2 compared with traced
reflections of all other years (grey) from this profile section. This is used to classify traced reflections in 2015 as reflections from the roof of
a water body (blue) and bedrock (red). The vertical exaggeration is 2-fold. **b:** The corresponding classification for 2015 posted on a TanDEM-
X DEM in September 2015.

Backscatter from the glacier bed, which at this stage can both be ice-bedrock and ice-water interfaces, is usually recognised as

the strongest continuous reflections in the 2D migrated amplitude images. The next steps including reflection tracing, sub-

sampling of traced reflections from 5 m interval to 20 m interval with filtering and masking of traced reflections near sharp

turns in profiles are the same as in Magnússon et al. (in review).

### 2.2 Outlining the lake margin

At this stage both the repeated migrated RES profiles as well as traced reflections were projected to common length axis of

the survey in 2014 (the 2018 survey for the new profiles measured since 2018) to allow direct comparison. The traced

reflections were first compared in areas at or outside the rim of ESC, undoubtedly also outside the subglacial lake margin. The

median elevation difference for the traced reflection in these areas, when compared to the master (2014), was used to bias

correct individual surveys in 2015–2020 towards the master, always resulting in <2.5 m vertical shift (in 2018 and later the

shift is obtained from comparison with an interpolated bedrock DEM based on surveys from previous years). At this stage the

comparison of the profiles (Fig. 2–3) reveals areas for which the elevation of the traced reflections (median corrected in 2015–

2020) is unchanged at the temporal minimum, between 2 or more survey dates, indicating reflections from bedrock for



corresponding surveys. The comparison also reveals areas where the traced reflection of a given survey is clearly above the traced reflection of another, designating a reflection from an elevated ice-water interface. This helps identifying the parts of a profile that are reflections from the lake roof (Fig.3). It also reveals that the edge of the lake is commonly characterised by relatively steep side walls, which further helps pinpointing the lake edge where repeated reflections from the bedrock were not

obtained, as in 2016 and 2019 when the lake area was at its smallest. The lake margin was then approximated in between the RES profiles to obtain the lake outlines and area (Fig. 4). Some of the RES profiles in 2014 and 2015 did not fully span the areal extent of the lake. The lowering during the 2015 jökulhlaup (see section 2.4) was therefore used to further guide this approximation of the 2015 lake margin where RES observations on the lake edge are not available. The obtained 2015 coverage and observed advance of the margin in 2014–2015 from the RES profiles was used to approximate the 2014 lake margin,

where this limitation applies to the 2014 survey. The drawing of the lake margin in 2016–2020 was however done based on the RES data alone. The south part of the lake margin in 2017 is a rough estimate (dotted line in Fig. 4d) since this part of the lake margin was beneath the snow covered supraglacial lake (Fig. 5), which corrupted the RES data obtained in this part of the cauldron. In 2018 the supraglacial lake was smaller and the margin of the subglacial lake had advanced beyond the extent of the supraglacial lake, hence the margin of the 2018 subglacial lake is expected to be quite accurate. Similar defects in the 2020

RES data (Fig. 2h), likely caused by englacial water bodies, made it impossible to detect part of the southern lake margin, and may result in a somewhat uncertain lake area. The approximated margin in 2020 (dotted line in Fig. 4g) is, however, constrained by traced reflections from bedrock a short distance south of the drawn margin, hence the estimated lake area in 2020 is near its expected upper limit.

**2.3 Creation of bedrock DEM and lake thickness maps**

The records of traced reflections were now split in two groups, using the lake outlines derived above: i) reflections from bedrock, and ii) reflections from the roof of the subglacial lake. The former data group, including data from profiles obtained in the vicinity of ESC (mostly in 2017 and 2019), were now merged into a single data set. The record of traced bedrock reflections covers fairly well the bedrock beneath the cauldrons except where the lake is present for all surveys (Fig. 4). In addition to the RES the bedrock elevation has been measured directly through two boreholes (Gaidos et al., 2020), which

fortunately were located within the RES data gap. From the bedrock record, including borehole measurements, a bedrock DEM (Fig. 6a) with 20m x 20m cell size has been interpolated using the kriging method (using a function in Surfer 13 © Golden Software, LLC).

The filtered and revised records of traced reflections from a given year obtained within the corresponding lake margin were assumed to originate from the lake roof at that time. An independent survey of the lake roof elevation was carried out through

a borehole close to one of the RES profile only 4 days after RES survey in June 2015 (Fig. 7), which supports our choice of $c_{gl}$ (see section 4.1.3).

EGU Open Access

**Figure 4 a–g:** Traced bed reflections (both ice-water and ice-bedrock reflections) for the RES surveys in 2014–2020. Locations of traced reflections of each survey are displayed in different colours on top of the survey route of each year (shown as grey lines). The contour map shows the surface elevation in September 2015 (TanDEM-X). Polygons (blue line) and numbers indicate derived margin and area of the subglacial lake for corresponding year. Poorly constrained sections of the lake margin are shown with a dotted-line. Locations of all traced reflections with corresponding colour-coding are shown in **h**. * One profile in 2015, surveyed by driving from the cauldrons centre out of study area towards northeast, was acquired in February 2015. It was only used to approximate the position of the lake margin in spring 2014 and 2015 and for tracing bedrock reflection outside the lake.



The lake roof records of individual survey epochs were now compared with the interpolated bedrock DEM to obtain lake thickness for each data point. Coordinate lists (20 m between points) of the determined lake outlines for each epoch with lake thickness set to zero were added to the corresponding RES lake thickness record before interpolating a lake thickness map (using the Kriging function in Surfer 13) for each year (Fig. 6). At a few locations minor adjustments of the interpolated maps

were made because of disagreement between crossing profiles. This only occurred in areas of very steep topography in the lake roof where 2D migration tends to fail, particularly for profiles driven perpendicular to the slope direction of the underlying lake roof (see section 4.1.2). In such cases the manual adjustment favoured data from profiles which were more parallel to the roof's slope direction. Lake volumes (Fig. 6) were obtained by integrating the individual thickness maps. In 2020, only the area could be obtained from the RES data; the lake topography was only partly surveyed (Fig. 6h) due to strong internal

reflections (see section 2.1), prohibiting direct integration of the lake volume. In this case, the volume of the lake was estimated assuming a linear relation between the lake area and volume using the values obtained in 2014–2019 (Fig. 6i).

## 2.4 Elevation changes and released volume of water during jökulhlaups

The DEMs used to measure the surface lowering of ESC during the jökulhlaup in 2015 were deduced from Interferometric Synthetic Aperture Radar (InSAR) data acquired by the TanDEM-X twin satellites on 23 September and 10 October, few days

before and approximately a week after the jökulhlaup. The DEMs are processed by extracting the topographic information from the InSAR data in the same manner as described by Rossi et al. (2012). The DEM difference reveals the area affected by the depletion of the subglacial lake as a clear anomaly, outlined in Fig. 8a, as well as the surface lowering above the flood route from the lake south of the cauldron. The DEM difference was corrected for actual near homogenous surface elevation changes between the two dates, unrelated to the jökulhlaup, and for slowly varying elevation errors in the DEMs, e.g. caused

by different penetration of the radar signal into the glacier surface at the two dates (Rossi et al., 2016) a correction was deployed. Around the outlined anomaly, excluding the flood route, ~500 m wide reference area was defined, where the elevation changes due to the 2015 jökulhlaup are expected to be insignificant (within few decimetres). The method of least squares was used to fit a linear plane through the obtained elevation difference within this reference area. This linear plane was subtracted from the elevation difference between the two DEMs.

The DEM prior to the jökulhlaup in early August 2018 was constructed from a DEM obtained as part of the ArcticDEM project (Porter at al., 2018) in August 2017, corrected with the DGNSS profiles acquired on 4 June, during the 2018 RES survey of ESC (Fig. 4e). The elevation changes, during the jökulhlaup, were obtained by comparing this DEM with the airborne radar altimetry profiles with approximate accuracy of 1–2 m (for more details see Gudmundsson et al., 2016), acquired on 9 August, a few days after the jökulhlaup (Fig. 8d). The difference between the DEM and the radar altimetry profiles were interpolated

with kriging to obtain a map of elevation changes during the jökulhlaup. To compensate for actual surface elevation changes from 4 June and 9 August, unrelated to the jökulhaup, a linear plane was again subtracted from the obtained map of elevation changes. The linear plane was obtained in the same way as for the jökulhlaup in 2015, except the westernmost part of the



reference area from 2015 was excluded, due to elevation changes related to a jökulhlaup from WSC, which occurred at the same time as the flood from ESC in 2018.

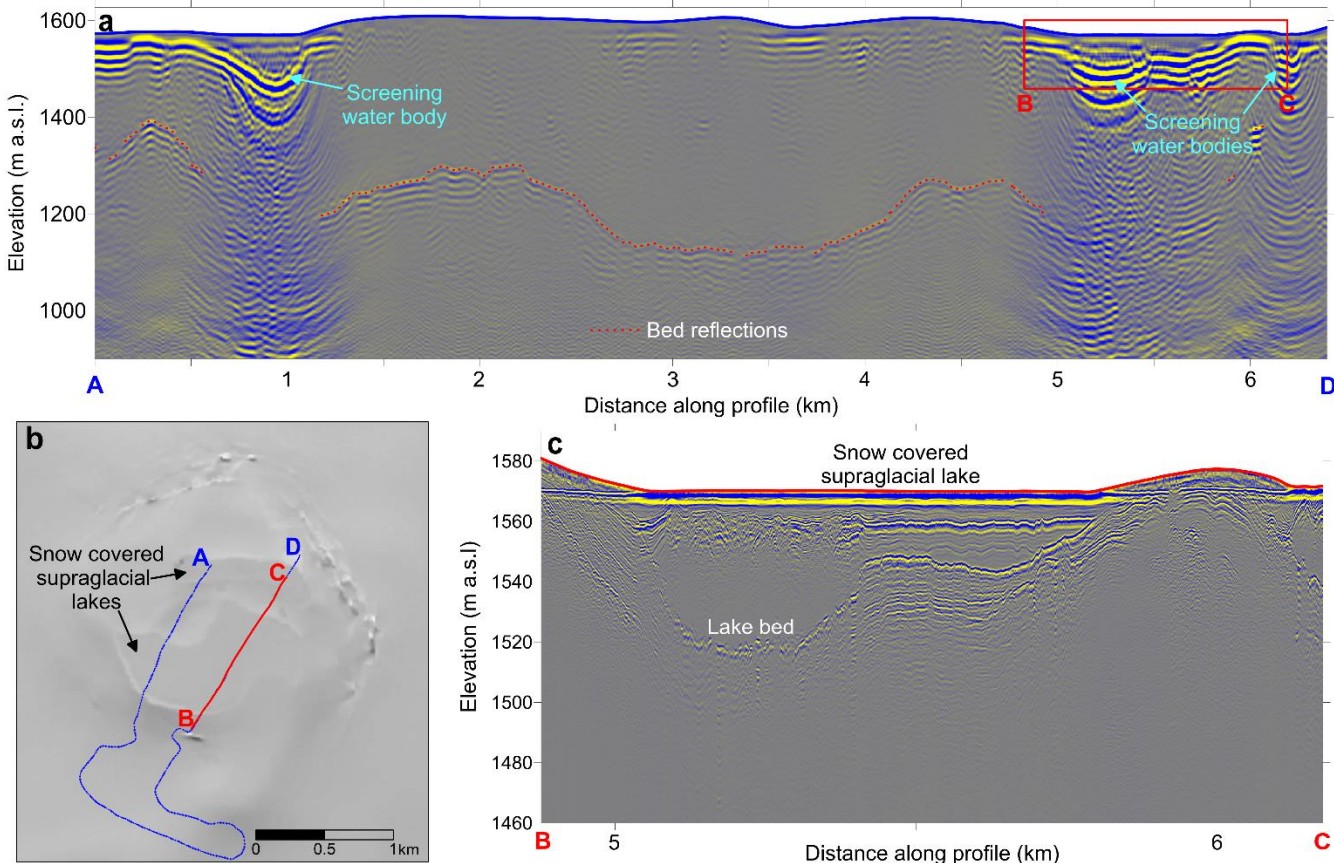

**Figure 5 a:** The low frequency (5 MHz) RES survey (2D migrated) on 7 June 2017 from location A to D (location shown in **b**) revealing features which induce ringing in the received radar reflections, completely screening reflections from the glacier bed (traced reflections indicated with a red dotted-line). The flat glacier surface above these features along with Landsat-8 optical image in August 2017 (**b**) clearly reveals these features as snow covered supraglacial lakes. RES survey on 8 June 2017 with 50 MHz Malå radar (**c**) along subsection B to C (location shown in **b**) repeating the low frequency survey (corresponding part of the low frequency RES-profile is indicated with red box in **a**) further confirms this. Note that the elevation projection for **c** is carried out using $c_{gl}$=1.68x10$^8$ m s$^{-1}$. The propagation velocity through the media above the supraglacial lake bed is much lower, hence the depth of the lake as indicated in **c** is overestimated. The vertical exaggeration is 2.5:1 and 5:1 in **a** and **c**, respectively.

To obtain a measurement of water volume released during the jökulhlaups, the elevation changes were integrated within the outlined area of lowering due to the depletion of the lake. The area where this lowering was more than few decimetres is quite distinctive in the 2015 elevation change map. The less accurate elevation change map during the 2018 jökulhlaup, due to the sparse altimetry data after the jökulhlaup (profile location shown in Fig. 8d) and a larger time gap between the pre-jökulhlaup DEM and the jökulhlaup (~2 months compared to only few days in 2015), made it difficult to directly outline the area of lowering in 2018. It was therefore assumed that the lowering area was the same as in 2015 (dashed line in Fig. 8c). The integrated volume change within this area was 280±5 Gl (10$^6$ m$^3$) for the jökulhlaup in 2015 and 180±18 Gl in 2018 (uncertainty





corresponds to a possible bias of 0.25 m and 1.0 m for the elevation change maps for in 2015 and 2018, respectively for the area of integration). To estimate the volume loss during the jökulhlaup, corresponding to the water released from the lake, we also need to consider the volume of crevasses formed during the jökulhlaups, which are not represented in the post-jökulhlaup elevation data. The crevasse field surrounding ESC after the jökulhlaup in 2015 formed an ~8 km long arc. Assuming that the

cumulative width of the crevasses across the 300–400 m wide crevasse field is 100 m at the surface, and that this width decreases linearly with depth to 0 m at 100 m depth results in a volume of 40 Gl (Guðmundsson et al., 2018). In 2018, the crevasse field had a similar area (shorter arc but wider) resulting in the same crevasse volume estimate. The uncertainties of these estimates are assumed to be rather high or 50% of the derived values. By adding the estimated crevasse volume, the estimated volume of water released from the subglacial lake increases to 320±20 Gl during the jökulhlaup in 2015 and 220±30

Gl in the 2018 jökulhlaup.

**3 Results**

The evolution of the lake area inferred from the RES surveys in 2014–2020 is shown in Fig. 4. The minimum areas of 0.5–0.6 km², was observed less than a year after the 2015 and 2018 jökulhlaups, while the maximum of 4.1 km² was observed in June 2015, ~4 months prior to a jökulhlaup. At the time of observed maximum, almost 5 years had passed from the previous

jökulhlaup from ESC in July 2010 (Guðmundsson et al., 2018). The lake had reached an area of 3.2 km² in June 2018, two months prior to a jökulhlaup.

The lake development in terms of volume and shape is shown in Fig. 6. It indicates rather strong relation between the area and volume of the lake (Fig. 6i). The variation of lake volume obtained with RES along with the estimated volumes of water released during the jökulhlaups extracted from surface elevation changes (Fig. 8c–d) are displayed in Fig. 9. The RES surveys

indicate lake volume <50 Gl in 2016 and 2019 and a maximum volume of 250 Gl is derived for June 2015 compared with a volume of 320±20 Gl released during the jökulhlaup ~4 months later. The survey in June 2018 yields a volume of 185 Gl, while the released volume in August same year was 220±30 Gl. The onset of the 2018 jökulhlaup was observed from real-time monitoring with a GNSS station operated in ESC (https://brunnur.vedur.is/gps/eskaftarketill.html) by IMO, few days prior to the peak discharge in the Skaftá river (Fig. 1). At the time of observed onset the water volume in the lake had already been

estimated to be 180 Gl, using the RES record from ESC in 2014–2018 (Guðmundsson, et al., 2018), indicating the applicability of our RES survey approach to evaluate the expected hazard from a jökulhlaup.

The development of the lake volume in 2010–2020 assuming it drained completely in the jökulhaup in July 2010, mimics a saw tooth curve (Fig. 9 a) with an approximately fixed filling rate of ~60 Gl a⁻¹ between jökulhlaups (Fig. 9 b). The values in 2014 and 2015 are slightly offset from this trend, possible due to less dense profile network then than for later surveys. This

may explain the better match in 2018 between the lake volume derived from the RES survey and the released volume few months later, compared with 2015. If the RES surveys of 2014 and 2015 are excluded the filling rate between jökulhlaups is ~65 Gl a⁻¹. It is worth noting how poorly the measured surface elevation at ESC centre correlates with the lake volume beneath







**Figure 6 a:** The location of traced reflections classified as reflections from bedrock (red lines) in the combined 2014–2020 RES record along with elevation of bedrock measured through boreholes (red triangles) used to interpolate a DEM of the bedrock beneath ESK and near vicinity. This DEM, represented with the elevation contour map (20 m contour interval), is shown in the background of **a–h**. **b–h:** Maps of lake thickness (colour bar below) along with location of traced reflections classified as reflections from the lake roof (red lines), used to interpolate lake thickness map for each survey. Lake volumes integrated from the lake thickness maps are displayed in Gl ($10^6$ m$^3$). **i:** The lake volume posted as function of lake area (in 2014–2019; black diamonds), which constrains linear relation (blue line) used to estimate the lake volume in 2020 (value marked with * in **h** and yellow diamond in **i**).



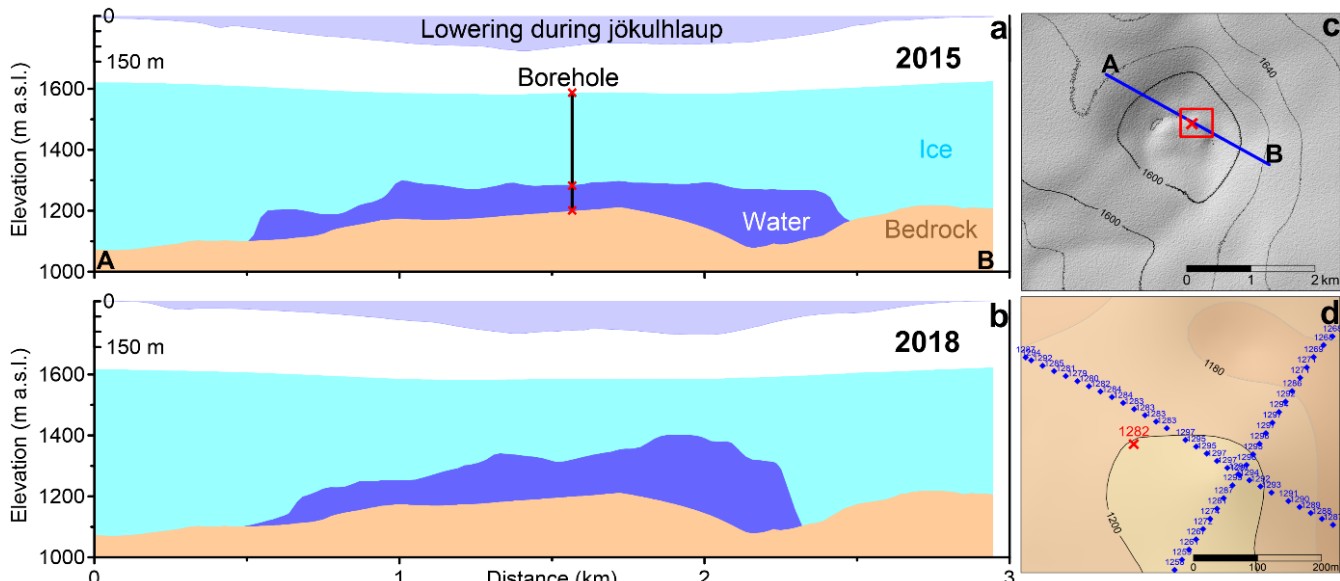

**Figure 7 a–b:** Cross-section over the centre part of ESC from location A to B (shown in **c**) revealing bedrock, lake and ice thickness, 4 and 2 months before the jökulhlaups in 2015 (**a**) and 2018 (**b**), respectively. The lowering along this cross-section during the subsequent jökulhlaup (derived from Fig. 8) is shown in the upper part of each panel. Note that the y-axis is without vertical exaggeration. **d:** Comparison of lake roof elevation measured with RES, 3 June 2015 (blue numbers and diamonds), and through borehole, 7 June 2015 (red number and x). The borehole location relative to the cross-section A to B is shown in **a** and **c**. Red box in **c** indicates the area shown in **d**.

ESC (Fig. 9). This indicates the governing role of ice dynamics for filling up the cauldron surface depression, while the contribution of water accumulation in the lake to surface elevation changes is small in comparison; large proportion of the accumulated water simply replaces ice melted beneath ESC.

A striking feature in the lake shape for all observations are steep side walls, clearly represented in Fig. 7a–b, typically exceeding 45° slopes and sometimes even 60°. Despite the apparent linear relation between the lake volume and area (Fig. 6i) the overall shape of the lake varies substantially during the study period. In 2014 and 2015, before the jökulhlaup in autumn 2015, the water was distributed much more evenly over the lake area than in 2018, when both the lake volume and area were close to the values obtained for 2014. In 2018, the lake water was, however, more concentrated close to the ESC centre with the maximum lake thickness above the crater-shaped bed depression beneath the eastern side of ESC (Fig. 6–8), ~0.5 km east of boreholes (Fig. 6a and 7a). The lake margin also appeared to be different in 2018. The steep side walls surrounded the bulk of the lake but between these walls and the outlined lake margin was an area with typical lake thickness of 10–30 m (see Fig. 6f and left side of Fig. 7b). This clear difference in the lake shape before the 2015 and 2018 jökulhlaups is also apparent in the lowering during these jökulhlaups (Fig. 8c–d). Despite greater lake thickness beneath the ESC centre in 2018 the surface elevation was similar as in 2015 (Fig 7a–b and Fig. 9a). The ice prior to the 2018 jökulhlaup above the lake was however relatively thin; in 2017 the minimum ice thickness was only ~150 m but it had increased to ~180 m in 2018. Prior to the 2015 jökulhlaup when the lake water was more evenly distributed the corresponding values were ~260 m and ~280 m in 2014 and 2015, respectively. The outward migration of the lake margin in 2014–2015 was characterised by outward propagation of the

steep side ice walls defining the lake margin by typically 50–150 m. The steep side walls also seem to characterize the lake

margin in 2017 but this was quite different in 2018. The steep side walls still surrounded the bulk of the lake and these walls had advanced from 2017 but the lake margin had advanced much further, typically 100–1000 m (Fig. 6e–f), due to the formation of previously mentioned 10–30 m thick water layer surrounding the steep lake walls.

**Figure 8 a–b:** The lake thickness maps for ESC (colour bar below) 4 and 2 months before the jökulhlaups in 2015 and 2018, respectively
(from Fig. 6c and 6f). **c–d:** Maps of glacier surface lowering (colour bar below) during these jökulhlaups with lake margin (cyan line) from **a** and **b**. The dashed red line indicates the area of notable surface lowering during the 2015 jökulhlaup. The grey lines in **d** indicate the locations of radar altimetry profiles surveyed from an airplane on 9 August 2018, a week after the jökulhlaup. The total volume of the lake integrated from the lake thickness maps (**a–b**) and the released volume integrated from the surface lowering during jökulhlaup adding estimated volume of crevasses (**c–d**) are displayed in Gl ($10^6$ m$^3$). **e–f:** The difference between lake thickness obtained by RES, in 2015 and
2018, and the lowering during the following jökulhlaup. Polygons filled with diagonal crosses indicate the areas of large crevasses formed during the jökulhlaups as outlined from Fig. 1b–c. The contour maps indicate surface elevation (20 m contour interval) from TanDEM-X, 10 October, 2015 (**e**) and from the altimetry profiles on 9 August 2018 (**f**) as explained in section 2.4. Green triangle in **a–f** indicate location of a GNSS station operating during both jökulhlaups.

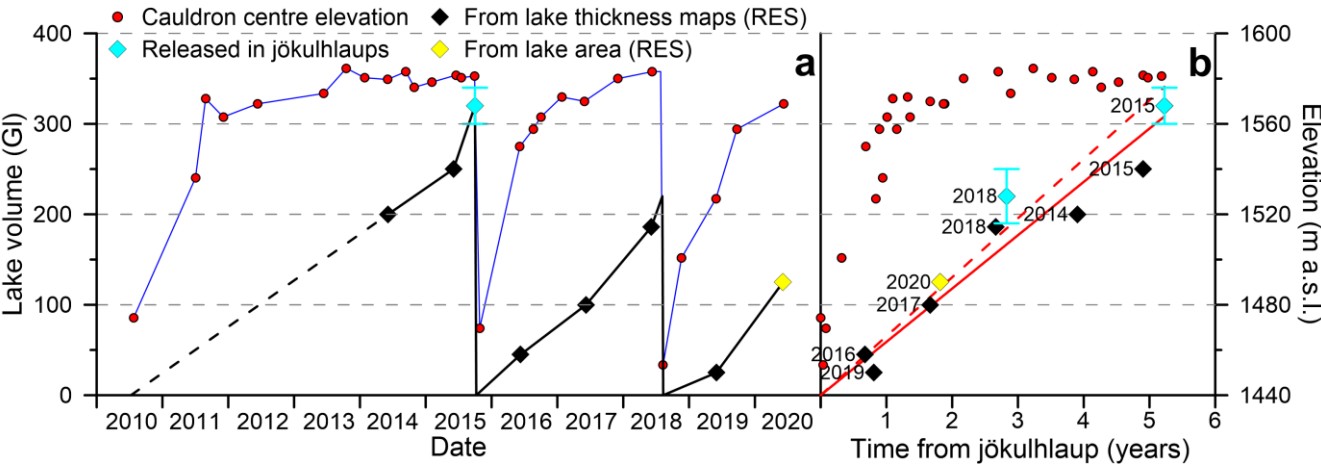

**Figure 9 a:** The development of the lake volume (left y-axis) beneath ESC in 2010 to 2020 obtained from the RES data (black and yellow diamonds) and derived surface lowering during jökulhlaups adding estimated volume of crevasses (cyan diamonds). The latter includes estimated uncertainty. It is assumed that the lake drained completely during the jökulhlaups in 2010, 2015 and 2018. Red dots show measured elevation (right y-axis) of ESC centre from radar altimetry and GNSS surface profiling (http://jardvis.hi.is/skaftarkatlar_yfirbord_og_vatnsstada). **b:** The same lake development and cauldrons centre elevation as function of time elapsed since previous jökulhlaup. The solid red line shows a linear fit through origin (zero volume at time zero) for the lake development; the dashed red line excludes the RES surveys in 2014–2015.

When comparing the obtained lake thickness map prior to jökulhlaups and the subsequent lowering (Fig. 7–8), the surveyed shape of the lake and the lowering shows strong similarities. The lowering appears like a spatially filtered version of the lake thickness shape, with the maxima at approximately same location and substantial lowering (>5 m) extending typically 200–500 m outside the lake margin as obtained from the RES survey (Fig. 8). Figure 8e–f shows the derived difference between the lake thickness in spring 2015 and 2018 and the lowering during the jökulhlaups a few months later, indicating where the ice became thinner or thicker during and shortly after the jökulhlaups, and the outlines of excessively crevassed areas formed during these floods. The main thinning areas as well as the main crevasse areas are located at or outside the main ice walls of the lake. In 2015, this coincides with the lake margin but not in 2018 as mentioned above. The main exception from this is the derived thinning in the northern part of ESC in 2015, which extends significantly into the cauldron. The lake thickness in this area is however, not covered with direct RES observation (red profiles in Fig. 8a), hence, the apparent thinning may be an artefact, as the relatively sparse RES profiling did not capture the amount of water stored in this area prior to the 2015 jökulhlaup. This further supports that the true lake volumes in 2014 and 2015, based on the RES data, is underestimated. The thickening areas approximately correspond to the lake roof within the ice wall of the lake and the surrounding crevasse fields formed during the jökulhlaups. The thickening in 2015 was wide-spread, typically less than 40 m and at the centre of the cauldron our estimation suggests thinning, but that may be due to scarce bedrock data at this location (Fig. 6a). In 2018, the thickening was much more localized and exceeded 40 m for substantial part of the area where the ice grew thicker. In both jökulhlaups, the area above the crater-like bed depression beneath at the eastern side of the cauldron yielded by far the greatest thickening. In 2015, the derived thickening at this location was up to 110 m, while in 2018 it was up to 170 m.



## 4 Discussions

### 4.1 The uncertainty and limitations of the RES survey for quantifying the lake development

In the results presented, we did not attempt to estimate the uncertainty of the lake volumes derived from the RES data specifically. Instead, the results derived from the RES data were validated by comparing them with the volume of water released during jökulhlaups, obtained from measured surface lowering. The good agreement between the RES surveys before the jökulhlaups in 2015 and 2018 and the volumes obtained from surface lowering during these jökulhlaups (Fig. 8–9) and how well the two datasets fit a linear relation with time elapsed since the previous jökulhlaup (Fig. 9b), suggest that the errors of the lake volumes derived from the RES data may typically be 10–20%, except when the lake is small and therefore not posing significant hazard. By measuring a denser RES profile network as done since 2018, the error has probably been lowered to ~10% when the surveying conditions are favourable. This can be considered low given the uncertainty factors that affect the RES surveys. These include uncertain value of $c_{gl}$, limitations of the 2D migration applied, and interpolation error due to sparse data coverage both for obtaining the bedrock DEM and the lake thickness maps. Each of these factors may produce systematic errors, which can lead to either an under- or overestimated lake volume. It is, however, possible that errors caused by different factors cancel out one another to some degree.

### 4.1.1 RES data gaps

The bedrock area concealed by the subglacial lake in all RES surveys is 0.35 km$^2$ or ~10% of the lake area in 2018 and less than that in 2015. The centre of this gap in the RES bedrock observations is constrained with direct observations of bedrock elevation through boreholes. The contribution of this bedrock data gap to errors in the lake volume estimates is therefore expected to be small, except when the lake is small and mostly within the area of limited bedrock data. At other locations in the RES profile network, reflections from the bedrock have generally been traced at some time point, meaning that for most observations on roof elevation there is also an observation on the bedrock elevation at the same location. Interpolation errors outside the bedrock RES data gap, contributing to errors in the lake volume estimate, are therefore mostly related to the interpolation of the lake thickness, not the bedrock elevation.

Supraglacial lakes and englacial water bodies, further discussed below, produce gaps in the data used to interpolate lake thickness maps for some years. For this reason, the lake volumes obtained in 2017 and 2020 could have errors as high as ~20%; in 2017 mostly due to uncertain location of the lake margin, in 2020 due to possible deviations from the obtained linear relation between lake volume and area (Fig. 6i). The survey in 2018 is also affected by similar data gaps. The lake margin is, however, fairly well constrained and only ~15% (~0.5 km$^2$) of the lake area (3.2 km$^2$) is affected by these data gaps. Interpolation errors in the lake thickness maps are probably resulting in larger lake volume errors in 2014 and 2015 when the distance between RES profiles was 400–500 m, compared with 200–250 m in 2018 and later.

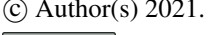



### 4.1.2 Limitations of the 2D migration

In most glaciological studies applying RES the acquired data only allows for 2D migration, which requires the assumption that all received radar reflections are only from features beneath the survey profile. Significant errors in locating reflectors with 2D migration due to this simplification typically arise in areas like the one studied here where the topography of the reflecting surfaces is highly variable. This is most pronounced when profiles are surveyed perpendicular to slope direction of the reflective surface but much smaller when the slope and profile directions are in parallel (e.g. Lapazaran et al., 2016). Due to this shortcoming the elevation of the traced reflective surface tends to be overestimated in the 2D migrated data. If the point in the reflective surface closest to the radar, corresponding to the traced reflection, is not directly beneath the RES profile but cross-track, the obtained ice thickness is underestimated and therefore the elevation of the mapped surface directly below the profile is too high. It is never too low unless the traced reflection does not corresponds to the closest spot of the inspected surface. An experiment comparing 2D and 3D migrated RES data obtained above steep bedrock beneath Gulkana Glacier, Alaska, clearly indicated such underestimate in bed elevation from the 2D migrated data (Moran et al., 2000). This was also illustrated in a recent study on Mýrdalsjökll ice cap (S-Iceland) using the same radar system as for ESC, where topographic settings and ice thickness are similar (Magnússon et al., in review). There, results from a topographic mapping of the bedrock from parallel profiles separated by 20 m, allowing 3D Kirchhoff migration (e.g. Schneider, 1978) of the RES data, were compared with a previous RES survey adopting 2D migration where majority of the profiles were driven in same direction ~200 m apart, in addition to less dense profiles with different survey directions. The results from the 2D migrated profiles were on average ~10 m higher than the bedrock DEM obtained from the 3D migrated data. In the study presented here, where crevasses and the size of the study area do not allow a safe acquisition of data for 3D migration with a reasonable effort, we may expect the 2D migration to introduce a similar bias. This, however, applies both to the reflections from the bedrock and the lake roof shifting both surfaces upwards, hence the effects of this may to a large extent be cancelled out, when estimating lake thickness and volume. This effect on the surveyed lake roof elevation should, however, vary between observations and be most prominent when the topography of the lake roof was most uneven in 2017-2018.

The RES profiles do not necessarily pass directly above subglacial topographic peaks, which may cause some further distortion in the lake thickness maps and bedrock DEM. In steep areas these topographic peaks are however represented as somewhat lower peaks at the RES profiles close to the actual peaks due to the cross-track reflection explained above. The height of topographic peaks in the lake may therefore be slightly underestimated and their exact planar position is likely somewhere between survey profiles but not directly beneath them as shown in Fig. 6b-h. The denser RES profile network surveyed since 2018 should reduce these errors.

### 4.1.3 Errors in radio wave velocity ($c_{gl}$)

We have a single borehole survey (Fig. 7d), which can be used to validate $c_{gl}$ used in the RES processing. The difference between the lake roof elevation at the borehole and nearest point on the profiles is 1 m when using $c_{gl} = 1.68 \times 10^8$ m s$^{-1}$. Taking





into the account the mismatch in profile and borehole location (~50 m) and the spatial variability in lake roof elevation from the RES data, it is unlikely that actual difference between the lake roof elevation at the two locations exceeds 10 m, setting a boundary on the $c_{gl}$ uncertainty, resulting in $c_{gl=}(1.68\pm0.05)\times10^8$ m s$^{-1}$. Further, $c_{gl}$ at this specific location and time may deviate

from the average value of $c_{gl}$ in the survey area. We consider it unlikely that $c_{gl}$ exceeds $c_{gl=}1.70\times10^8$ m s$^{-1}$, corresponding to propagation velocity through dry ice with density 900 kg m$^{-3}$ (Robin et al., 1969). The water content in the temperate ice can, however, reduce $c_{gl}$ significantly (e.g. Smith and Evans, 1972), even below 1.60 $\times10^8$ m s$^{-1}$ (e.g. Murray et al., 2000). Given the value obtained at the borehole, we consider it unlikely that the average value for the survey area corresponds to $c_{gl}$ $<1.60\times10^8$ m s$^{-1}$. If we assume that the spatially averaged value of $c_{gl}$ is approximately the same for all surveys (as suggested

by the good comparison of repeated bedrock profile sections), the error in $c_{gl}$ should shift both the lake roof at all times and bedrock in same direction proportional to the ice thickness (without a lake above in case of bedrock) except for the relatively small part of the bedrock DEM constrained by borehole measurements (Fig. 6a). Consequently, the error in lake thickness as well as volume due to erroneous $c_{gl}$ should be proportional to the error in the applied value of $c_{gl}$. If too high $c_{gl}$ is used, the lake thickness is overestimated and underestimated if $c_{gl}$ is too low. Using $c_{gl=}1.68\times10^8$ m s$^{-1}$ if correct value is only

$c_{gl=}1.60\times10^8$ m s$^{-1}$ would lead to ~5% overestimate on the lake volume and considering the possible upper limit, $c_{gl=}1.70\times10^8$ m s$^{-1}$, significant overestimate due to wrong value of $c_{gl}$ is unlikely. Some of the errors introduced by using too high $c_{gl}$ may be cancelled out by the 2D migration tending to shift reflective surfaces upwards as explained above. For too low $c_{gl}$ the 2D migration may further exaggerate these errors.

### 4.1.4 Supraglacial lakes and englacial water bodies

Due to the temporary presence of supraglacial lakes within ESC (Fig. 5) and englacial water bodies beneath it (Fig. 2h) the value of $c_{gl}$, may differ significantly between some lake roof measurements and bedrock measurements at some locations. This may lead to more uncertain lake thickness than described above, assuming temporally constant $c_{gl}$. Supraglacial lakes sometimes form within ESC, probably as a consequence of highly compressive strain rates at the cauldron centre sealing water routes from the glacier surface down to the subglacial lake, resulting in accumulation of surface melt water within the cauldron.

It is worth noting that it is possible to trace in 50 MHz radar data (Fig. 5c) a flat water table of an aquifer layer extending from and between the supraglacial lakes. The presence of a supraglacial lake both screens out reflections from the bed beneath the supraglacial lake and reduces $c_{gl}$, due to increased water content in the media penetrated by the radar. This may affect the traced reflection in areas where the supraglacial lake is not deep enough to fully screen out reflections from the bed or due to high water content close to the glacier surface related to an aquifer layer. This effect was observed in the 2017 RES survey.

Then bed reflections outside the subglacial lake, at the edge of the supraglacial lake, appeared up to 20 m below the bedrock elevation observed at same locations in 2019. The lower elevation of the 2017 reflection was attributed to delay caused by the supraglacial lake and therefore not traced. Around 100 m farther away from the supraglacial lake in 2017, the RES surveys in 2017 and 2019 showed the bed reflections at approximately the same elevation indicating that a delay caused by the aquifer layer extending from the lake in 2017 is insignificant or limited to the shore of the supraglacial lake. The delay caused by a



shallow supraglacial lake may result in 10–20 m overestimate in the depth of some of the traced reflections in 2017 and 2018 near the data gaps seen as grey (untraced) profiles near ESC centre in Fig. 4d–e. This may contribute to a corresponding underestimate of the lake thickness for minority of the traced reflections from the lake roof in 2017 and 2018. It is worth noting that the unusually hilly topography of the lake for the same years is likely to cause unusually high upward shift of the lake roof elevation through the previously described limitation of the 2D migration, contributing to an overestimate in lake thickness. It

is not certain which of these two counteracting errors influence the derived lake volumes more in 2017 and 2018.

In 2020, englacial features screen out reflection from the bed (Fig. 2h) in the same way as the supraglacial lakes in 2017 and 2018. There were no indications in 2020 of snow covered supraglacial lakes and these features appeared at greater depth than in 2017 and 2018, hence these artefacts in 2020 are attributed to an englacial water layer (sill). Such layers probably need to be several metres thick to produce similar artefacts as the supraglacial lakes, which was apparently the case for a large part of

the ESC centre area in 2020. As a result, reflections from the lake roof could only be traced for a minority of the profiles crossing the subglacial lake. Fortunately, the lake margin could be mapped allowing an estimate of lake volume, due to the previously mentioned strong relation between the lake volume and area in 2014–2019 (Fig. 6i). When viewing the RES profiles for other years (Fig. 2), we typically see englacial features likely related to water bodies or layers, too thin to screen reflections from the lake roof and the bedrock. There are even indications of such a layer near the centre of the RES profile in Fig. 2c

corresponding to the time (2015) and the location where the lake roof elevation was directly measured through a borehole (Fig. 7d), showing matching lake roof elevation with $c_{gl}$=(1.68±0.05)x10$^8$ m s$^{-1}$. This indicates that despite likely existence of these englacial water bodies they are not causing an excessive delay, and likely affecting all RES surveys in a similar manner in 2014–2019. Likely deviation of $c_{gl}$ in 2020, due to thick englacial water layers, does not affect the corresponding lake volume, as it was estimated using the derived lake area, not by integrating a lake thickness map.

The RES surveys in 2014–2020 have revealed supraglacial lakes as temporal features sometimes forming in ESC and even though englacial water bodies and layers are generally found beneath the cauldron, it seems that in 2020 these features were more prominent than in other years. This highlights the temporal variability in the englacial and supraglacial hydrology at or beneath ESC. As suggested by Gaidos et al. (2020), the englacial water bodies may play an important role in the triggering of jökulhlaups from the Skaftá cauldrons. A jökulhlaup from WSC in 2015 was most likely triggered via the drilling of a borehole

at the cauldron centre, which created a pressure connection between the subglacial lake and an englacial water body above it (Gaidos et al., 2020). Sudden drainage of supraglacial lakes down to the glacier bed (e.g. Das et al., 2008) also highlights these lakes as a potential trigger of jökulhlaups from subglacial lakes, which should be studied further.

## 4.2 The shape of the subglacial lake and its evolution in 2014–2020

The repeated RES surveys in 2014–2020 yield new insight into the shape of the subglacial lake beneath ESC and how it has

evolved in recent years. The steep, almost step like, side walls (Fig. 2, 6 and 7) differ from the typical conceptual models of lakes beneath ice cauldrons (e.g. Björnsson, 1988; Einarsson et al., 2017) with the lakes drawn with smooth, approximately parabolic or elliptic, cross-sections. It is also different in form from attempts to approximate the lake shape based on the





with much lower melt rate on the lower part of the ice walls. To fully explain the observed shape of the lake will, however, require extensive modelling that is beyond the scope of this paper. Such a model should ideally combine ice dynamics, the pressure and thermal regime of the lake, the heat exchange between the lake water and ice walls as well as the interaction of the geothermal area with the lake, the floating ice above the lake and the surrounding glacier. A model of such complexity may also be required to fully explain the difference in lake shape before the 2015 and 2018 jökulhlaups (see section 3). Part

of the explanation is however probably related to changes in the geothermal area below ESC. Temperature profile measurements in the subglacial lakes beneath the Skaftá cauldrons have revealed stable temperatures with depth of about 3.5–5°C, allowing effective convection to take place (Jóhannesson et al., 2007; unpublished data at the IMO), and chemical analyses of the water in WSC lake revealed a component of geothermal fluid of deep origin at ~300°C (Jóhannesson et al., 2007). In 2016–2018 the main vents of the geothermal area, forming centres of a strong convection plumes with peak basal

melt directly above, were probably close to the two main maxima in lake thickness observed in all three years at approximately the same location (Fig. 6d–f). These maxima, indicating the locations where most ice had been replaced by meltwater since the 2015 jökulhlaup, were beneath the east side of the cauldron, above the west side of a sharp crater-like depression in the bedrock (section 3) and ~800 m farther west, close to the cauldron centre. The same maxima had started forming in 2019 (Fig. 6g) and at least the eastern one had continued growing in 2020 (Fig. 6h). During the period 2010–2015 these two vents in the

geothermal system were probably not as powerful as in 2015–2018, explaining the large difference in minimum ice cover thickness for these two periods (260–280 m in 2014–2015 vs. 150–180 m in 2016–2018). A substantial part of the geothermal power in 2010–2015 was likely released by other parts of the geothermal area beneath ESC, which typically are much weaker or dormant, explaining the relatively uniform lake thickness in 2014 and 2015. A temporal increase in geothermal activity in 2010–2015 probably occurred near the northern- and southernmost part of the lake in 2014 and 2015. The observed lowering

during the jökulhlaup from ESC in 2010 and the evolution of the ESC since the mid 20[th] century (Gudmundsson et al., 2018) indicates that this behaviour in 2010–2015 was unusual for the geothermal area and the activity in 2015–2018 resembles more the behaviour prior to 2010. Even though the distribution of the released geothermal energy was different for the two periods the net power of the geothermal area was probably similar, as represented in a similar rate of water accumulation in the lake over time (Fig. 9b).

Despite the indication of changes in the geothermal area, it should be kept in mind that the lake accumulated water for five years before the jökulhlaup in 2015 compared with three years for the 2018 flood. Some of the difference in lake shape may be due to this. However, the thickening of the ice cover in 2017–2018 (~30 m a$^{-1}$ at the cauldron centre) and the outward migration of steep ice walls seems too slow to explain the different lake appearance in 2015 compared with 2018. The difference in lake shape may, however, have contributed to the earlier onset of the jökulhlaup in 2018. The shallow lake area

outside the steep ice walls in 2018 may be an indication that the glacier outside the walls had started to float up as a consequence of high water pressure in the in subglacial lake. This high subglacial water pressure likely extended somewhat away from the



lake through connections in the subglacial drainage system outside of the lake. This may have contributed to the onset of a jökulhlaup two months later.

## 4.3 The jökulhlaups in 2015 and 2018

The jökulhlaup in 2015 has been the subject of recently published studies. Ultee et al. (2020) estimated the tensile strength of the glacial ice from the location of crevasse fields formed during the jökulhaup and Eibl et al. (2020) studied the seismic tremor related to the jökulhlaup and the potential of using seismic array measurements of the tremor for early-warning of subglacial floods. The jökulhlaup in 2018 has not yet received similar attention. The on-line GNSS station, operated by IMO, was running at approximately the same location near the centre of ESC (Fig. 8) during both jökulhlaups. The motion of this station shed an

interesting light on the course of events during the jökulhlaups considering the RES measurements of lake shape before the jökulhlaups and the surface lowering during them (Fig. 10).

During the weeks prior to the jökulhlaup the station had been rising relatively fast likely due to rapid inflow melt water from the glacier surface. The rate of uplift was ~0.12 m d$^{-1}$ and ~0.16 m d$^{-1}$ the last days before the jökulhlaups in 2015 and 2018, respectively. This may be due to similar rate of inflow; the ~30% larger floating ice cover attributes to lower uplift rate in

2015. The start of the jökulhaups were observed as the end of these uplift periods in the late evening of 26 September 2015 and 1 August 2018. The start of the jökulhlaup was substantially slower in 2015. The station subsided by ~2 m during the first day of the jökulhlaup in 2018, while in 2015 it took almost 3 days to reach a similar subsidence (Fig. 10). The slower initial subsidence in 2015 can only partly be due to difference in lake area; this should only explain similar difference as observed in the uplift prior to the jökulhlaups.

After 2 m subsidence, the GNSS station dropped by 60 m in 2015 and 81 m in 2018 over a period of ~40 hours. Then, ~4.7 and ~2.7 days into the jökulhlaup in 2015 and 2018, respectively (times marked with circles in Fig. 10a), the station subsidence started to decelerate and at the same time an eastward motion started. This was followed by a period of decelerated subsidence lasting for ~7 hours in 2015. This period probably corresponds to the time when a "keel" at the bottom of the floating ice cover clashes with the bedrock beneath or close to the station. The net subsidence of 68 m at the end of this period (marked with

grey triangle in Fig. 10a) fits well with the 67 m lake thickness obtained at the GNSS station as the difference between the bedrock DEM (the GNSS station was located less than 80 m from boreholes where the bedrock elevation was measured directly) and the traced lake roof elevation in June 2015. In 2018 the period of decelerating subsidence lasted for a day. The 94.5 m net subsidence by the end of this period (marked with black triangle in Fig. 10a) is substantially less than the 140 m lake thickness obtained 50 m north of the station in 2018. This lake thickness is, however, obtained at the side of a steep up-

doming of the lake roof. The traced lake roof elevation at this location in 2018, was therefore sensitive to the limitation of the 2D migration (section 4.1.2) and likely corresponds to a reflection from the lake roof 100–200 m farther NNE (traced reflection from RES profile surveyed with ESE direction).

It is worth noting that during the main subsidence period in 2018, a sudden temporal deceleration occurred in the subsidence as well as in ice flow direction after only 15 m subsidence ~1.6 days into the jökulhlaup (marked with star in Fig. 10a). Such



a deceleration is not observed in 2015. It may be related to the 10–30 m thick water layer observed around the main water

chamber in 2018. After a subsidence of only 15 m this floating ice may have hit bedrock only a few hundred metres south of

the GNSS station. Due to the supraglacial lake, the closest traced reflections in June 2018 south of the station were 450 m

away. They indicate grounded ice or lake roof only few metres above the bedrock.

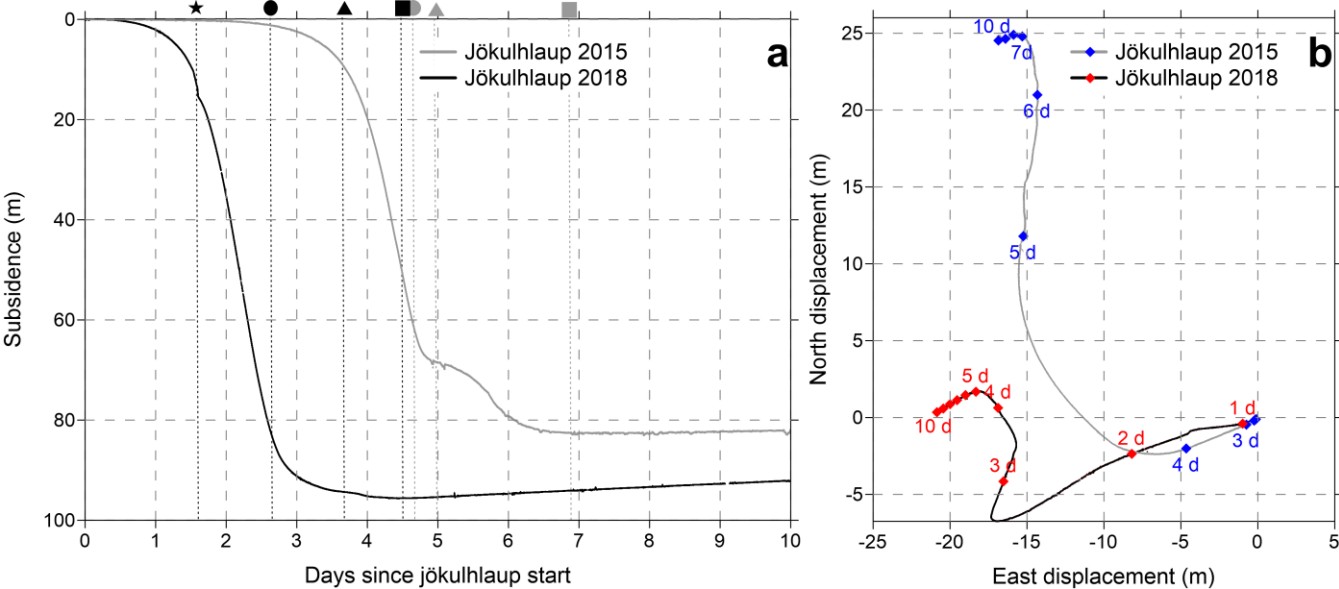

**Figure 10 a:** The subsidence of the GNSS station in ESC (exact location shown in Fig. 8) during the jökulhlaups in 2015 (grey profile) and
2018 (black). The times marked with forms (star, circles, triangles and squares) are discussed in the section 4.3. **b:** A planar view showing
the horizontal track of the station during the jökulhlaup relative to its position at the onset of the jökulhlaup. Blue and red diamonds show
positions of the station at 24 h intervals during the 2015 and 2018 jökulhlaups, respectively.

After the period of decelerating subsidence in the late stage of the jökulhlaups, the subsidence temporally sped up again in

both jökulhlaups. The speed-up was quite significant in 2015 but only minor in 2018. The station reached a total subsidence

of 82.6 m in ~6.9 days during the 2015 jökulhlaup (grey square in Fig. 10a) and 95.6 m in ~4.5 days, three years later (black

square in Fig. 10a). The horizontal motion of the station continued to decelerate and change direction for a bit more than a day

during both jökulhlaups. This probably marks the jökulhlaup terminations ~8 and ~6 days after they started in 2015 and 2018,

respectively. Lake water was probably still draining slowly from beneath the areas where the lake was thickest both in 2015

and 2018, east and west of the GNSS station, beyond the period of subsidence as recorded by the GNSS station, during the

period of gradual slow-down in horizontal motion. At the end of the 2015 jökulhlaup, the station was located on a relatively

steep northward sloping glacier surface (Fig. 8e). The lowering during the final phase of this jökulhlaup, when the station is

moving rapidly in north direction (Fig. 10b), is therefore, to some extent ice motion parallel to the glacier surface slope. The

station lowered by 15.5 m and moved by similar distance northwards during this period. The ice surface geometry near the

station near in the late stage of the jökulhlaup may favour local thinning due to strong tensile strain rates, which may also

partly explain the net thinning of the ice obtained near the station in 2015 (Fig. 8e). In 2018, the GNSS station ended at a

relatively flat area, resulting in much less subsidence and horizontal motion during the final phase of the jökulhlaup.



The motion of the GNSS station during the jökulhlaups gives insight into the scale of the events in terms of ice movements, which further helps understanding the difference between obtained lake thickness prior to the jökulhlaups and the surface lowering during the jökulhlaups (Fig. 8). In addition to the subsidence >70 m in a single day in 2018 (>50 m in 2015), the maximum horizontal velocity of the station was above 10 m d$^{-1}$ in 2018 and around 20 m d$^{-1}$ in 2015. The net horizontal displacement during the jökulhlaup, which did not follow a straight line, was approximately 30 m in 2015 and 20 m in 2018 (Fig. 10b). We may expect that the horizontal displacement at the location of the station at the cauldron centre is substantially less than near the sides of the cauldron where the ice-flux towards the cauldron centre is highest. There the net horizontal displacement may exceed 100 m. With this in mind, it is easier to understand how thickening of ice at given location may be up to 170 m as estimated in 2018 (Fig. 8f). The 100–200 m high walls of the main water chamber in 2018 with slopes sometimes exceeding 60° (Fig. 7b) possibly moving many tens of metres inwards, may therefore produce >100 m increase in apparent ice thickness near the pre-jökulhlaup ice walls. The extension of the thickening area into the main crevasse field at the north side of ESC in 2018 (Fig. 8f) is probably an expression of ice dynamics of this kind. Even though the ice in this area became thicker it suffered high tensile strain rates causing the crevasse formation. This effect is, however, expected to be largest in the east side of the cauldron where the estimated ice thickening is by far greatest (Fig. 8e–f). In this area, we observe the steepest and highest ice walls of the lake prior to the jökulhlaups, particularly in 2018. This was also the area surrounded with the largest crevasses in 2018 (Fig. 1c). Additionally, the bedrock at this location is steeply inclined towards a deep bedrock depression beneath the thickest part of the lake (Fig. 6–7). This may enhance sliding of the ice towards the depression centre during the jökulhlaup; inward sliding of the ice walls would produce stronger apparent thickening than if these ice walls would only be tilted inwards without sliding along the bed.

When the net inward horizontal motion may decrease from ~100 m to zero over a distance of few hundred meters, we may expect that thickening of the ice caused by compressional straining during the jökulhlaups was several tens of meters, comparable with the ice thinning observed outside the lake (Fig. 8e–f) by tensile straining. The high compressional strain rates are evident in compressional ridges that are formed near the centre of the cauldron during jökulhlaups (Fig. 1e) as well as the high uplift rate of the GNSS station after the jökulhlaups. In 2018, the uplift rate of the station the first days after the jökulhlaup was ~0.7 m d$^{-1}$ (Fig. 10a). When the surface elevation of the cauldron was mapped on 9 August, the station had risen by almost 3 m from its lowest elevation (on 5 August), likely due to post-jökulhlaup ice thickening caused by compressional straining. The post-jökulhlaup strain rates are expected to be much lower than during the jökulhlaups; the horizontal velocity of the GNSS station during them was an order of magnitude higher than after they ended.

The data sets obtained during the jökulhlaups in 2015 and 2018, could be further used to extract information about the mechanical properties of glacial ice, such as parameters describing viscous and elastic deformation and fracture strength. Interpretation of the available data about ice-surface lowering and the geometry of the ice shelf and subglacial water body in terms of mechanical properties requires the coupled modelling of the dynamics of the ice shelf and outflow from and the water pressure in the subglacial lake. For modelling the collapse of the cauldron during these jökulhlaups, the RES observations define the shape of the lake at the start of drainage and the subsidence of the GNSS station can be used as a constraint on the



water outflow from the lake during the jökulhlaup. The time-dependent pressure in the lake is required as a boundary condition to describe to what extent the weight of the overlying ice is supported by stresses in the ice and to what extent the ice floats on the subglacial water body. The result of such a modelling experiment, mimicking the observed elevation changes and crevasse

formation, may advance the modelling of ice dynamics during extreme strain rates, such as for glacier calving. Such a model, which may require a particle-based model of glacier dynamics to fully include the brittle behaviour of the glacier ice (Åström et al., 2013), could also be used to estimate temporal variations in the lake water pressure during the jökulhlaup. This might answer whether sudden temporary drops of water pressure in the lake may trigger lowering of pressure within the uppermost part of the geothermal system beneath the ESC, considered as an explanation of powerful low frequency seismic tremor pulses

(Eibl et al., 2020; Guðmundsson et al., 2013b) that have often been observed near the end of jökulhlaups from the Skaftá cauldrons.

## 5 Conclusions

The RES data presented from the Eastern Skaftá cauldron and a comparison with surface lowering during jökulhlaups, yielding independent measurements on the lake volume, shows that RES can be used for quantitative monitoring of the lake volume.

No other type of measurements have provided such volume estimates for ESC prior to jökulhaups, a key knowledge for assessing the hazard of a potential jökulhlaup. The study presents new insight into the shape and the development of a subglacial lake beneath ice cauldrons, maintained by subglacial geothermal activity, as well as the complex hydrological systems related to these cauldrons, not only beneath the ice but also within it and at its surface. In addition, the study provides a unique view on how the shape of a subglacial lake beneath ice cauldrons is reflected in lowering of their surface during

jökulhlaups. These new observations therefore provide interesting study opportunities related to ice cauldrons, including e.g. studies on:

    i)      The interaction between the geothermal area, the lake and the ice, as reflected in the shape and development of the lake.

ii)     The triggering mechanism of jökulhlaups from lakes beneath ice cauldrons.
    iii)    The ice dynamics and processes taking place within and beneath ice cauldrons during large jökulhlaups.

## Data and code availability

The authors declare that all code and data presented in the paper are available upon request except the TanDEM-X data provided by DLR, which is restricted to the users defined by the project NTI_BIST6868.
## Author contributions

EM and FP designed the research study and methods, carried out the all the low frequency RES-surveys as well as all processing related to these surveys. MTG and ThH were responsible for the survey and processing of the radar altimetry data acquired in August 2018. CR processed the TanDEM-X DEMs used in the study. BGÓ and TJ were responsible for operating the GNSS stations during the jökulhlaups in 2015 and 2018. ThTh lead the drilling into the subglacial lake beneath ESK in June 2015 and ES was responsible for the survey with the 50 MHz radar in June 2017. EM made all figures except Figure 7, which TH designed. EM prepared the manuscript with contributions from all co-authors.

## Competing interests

The authors declare that they have no conflict of interest.

## Acknowledgements

This work was funded by the Icelandic Research fund of Rannís within the project Katla Kalda (project nr. 163391) and the Icelandic Avalanche and Landslide Fund through the volcanic hazard assessment program GOSVÁ. TanDEM-X data was provided by DLR through the project NTI_BIST6868. Landsat-8 image courtesy of the U.S. Geological Survey. Copernicus Sentinel-2 data from 2018 was processed by ESA. All the RES-surveys were carried out during the annual field trips of the Iceland Glaciological Society (JÖRFÍ) on Vatnajökull. Sveinbjörn Steinþórsson, Ágúst Þór Gunnlaugsson, Vilhjálmur S. Kjartansson, Bergur H. Bergsson and Bergur Einarsson as well as JÖRFÍ volunteers are thanked for their work during field trips.

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
