# Peer review of "Development of a subglacial lake monitored with radio-echo sounding: Case study from the Eastern Skaftá Cauldron in the Vatnajökull ice cap, Iceland"

_The Cryosphere, 2021_

## Referee Comment (RC1)

**Review of Manuscript TC-2021-65:** *Development of a subglacial lake monitored with radio-echo sounding: Case study from the Eastern Skaftá Cauldron in the Vatnajökull ice cap, Iceland*

**General Comments**

This paper presents a new technique to monitor subglacial lake activity in Iceland using repeat Radio Echo Sounding (RES) surveys. The authors demonstrate the applicability of the new technique by deriving lake area and volume estimates between 2014 and 2020 for the subglacial lake below the Eastern Skaftá Cauldron (ESC), Vatnajökull. They then derive surface elevation changes from InSAR and airborne radar altimetry and compare these to the lake volume estimates to characterise two jökulhlaups which occurred in 2015 and 2018. Detailed long-term measurements of subglacial lake activity are rare but vital in understanding hazards posed by glacio-volcanic activity (e.g. jökulhlaups). The underpinning methods appear robust with some clarification required in placed and I've detailed these in the Technical Corrections below. In this context, the study is unique and worthy of publication in The Cryosphere, but subject to some revisions detailed below.

**Specific Comments**

The paper as it stands is quite long, stretching over 30 pages, and can be shortened – I've tried to provide suggestions in my Technical Corrections below to help the authors achieve this. For example, the Discussion section is 8 pages long and can be condensed by writing in a more concise manner. In particular, the authors sometimes discuss topics that are not relevant to a particular section and thus break up the flow of the discussion. For example, the discussion of uncertainties in the RES data pertaining to supra- and englacial water bodies is followed by a discussion on glacier hydrology and how this impacts jökulhlaup processes. These items should be discussed separately. Other suggested changes to the text are: breaking up the Results section into sub-sections and moving the discussion of RES errors into the methods section. In general, by ensuring each section is more focussed on developing the discussion pertaining to a particular section, each section will likely reduce in size and lead to a more concise paper overall.

Further, the writing in places is vague. The authors often describe the results with words such as 'seemingly' and 'almost' without providing quantitative statements from the results to back up their claims. In general, this can be resolved with a few simple calculations, but in other places the authors should consider referencing other suitable data sets or studies. This is particularly true in the Conclusion section which is written as more of a summary. Here, it would be better to provide some of the headline results from the study, backed up by quantitative statements, and state what the key take-home messages are from this study. I think the two main conclusions are:

- A new method has been developed for long-term monitoring of a subglacial lake which helps improve understanding of jökulhlaups.
- A unique analysis of jökulhlaups in 2015 and 2018 has been conducted that uses estimates of lake volume and surface elevation changes to understand the lake development before the jökulhlaups.

The methods underpinning this study are generally robust. However, the estimation of lake volume changes is dependent on having an accurate bedrock DEM. Is it possible that the bedrock outside and below the lake has changed over the course of the study period due to subglacial volcanic activity? One way in which you could quantify this is to find overlapping RES measurements representing the bedrock beneath the lake and calculate the vertical deviation between surveys. You have done this for areas outside the lake, but it might be possible to look at this below the lake.

Overall, if the authors can incorporate these suggestions and those in the Technical Corrections below, I'm sure this will make a very nice paper.

**Technical Corrections (References to line numbers in preprint)**

*Abstract*

L13: "The ESC is a ~3 km"

L14: "subglacial lake"

L15: "summer"

*Introduction*

L26-27: "Subglacial lakes have been directly and indirectly observed beneath both temperate and cold-based glaciers. The sudden release of water from such lakes can lead to floods, commonly referred to as jökulhlaups, and can be of variable magnitude.". Reference?

L27-28: "In warm-bedded glaciers, jökulhlaups are known to enhance basal sliding and increase glacier flow over a period of days (e.g. Einarsson et al., 2016)"

L32: I suggest beginning this paragraph with a short sentence summarising how subglacial lakes have previously been observed before launching into the paragraph, e.g. "Documenting the existence of subglacial lakes has been achieved using a combination of radio-echo sounding (RES) data and satellite remote sensing, but routine monitoring of such lakes remains a difficult task.".

L34: The exact quotation here should be "thick water layer beneath the ice".

L34: "Since then,"

L35: "RES data"

L35-39: Suggest combining these sentences together: "However, many subglacial lakes actively drain and fill and as a result are difficult to distinguish in RES data (Carter et al., 2007; Siegert et al., 2014), hence SAR interferometry and repeat altimeter surveys have been used to identify hundreds of areas of surface elevation changes associated with active subglacial lakes in Antarctica (e.g. Gray et al., 2005; Smith et al., 2009)."

L40-47: Suggest combine with paragraph below and shorten. See next few comments.

L40-44: Suggested start of new paragraph: "In Iceland, subglacial lake drainage events that lead to jökulhlaups have been documented since the early 1900s (Thorarinsson and Sigurðsson, 1947; Thorarinsson, 1957) and are well known to cause widespread destruction of farms and infrastructure, as well as loss of life."

L44-47: "Subglacial lakes in Iceland are formed through localized geothermal activity, where enhanced basal melting forms topographical depressions on the glacier surface (ice cauldrons), creating a low in the hydrostatic potential and promotes water accumulation from both the glacier surface and bed (Björnsson, 1988).

L49: "the two skaftá cauldrons, denoted as the Eastern Skaftá Cauldron (ESC) and the Western Skaftá Cauldron (WSC).".

L50: Change "within" to "below".

L50: "and has historically been the source of large jökulhlaups that have drained from beneath the Skeiðarárjökull outlet glacier in Southern Vatnajökull (Wadell 1920)."

L53-56: Suggest combining these sentences together: "The first direct observation of the ESC was from a photograph taken in 1938 whilst the WSC was first observed in 1960 (Guðmundsson et al., 2018), suggesting the ESC and WSC have not always co-existed.".

L56-61: "The geothermal power beneath Grímsvötn has been estimated from the volume of water discharged through jökulhlaups and surface mass balance and is estimated to be approximately 1500-2000 MW (Björnsson, 1988; Björnsson and Guðmundsson, 1993; Guðmundsson et al., 2018; Reynolds et al., 2018; Jóhannesson et al., 2020). This is likely to be similar below both the ESC and WSC making the region some of the most powerful geothermal areas in Iceland.".

L62: "which resulted in 4.7 $km^3$ and 3.4 $km^3$ of water being released, respectively (Gudmundsson et al., 1995; Björnsson, 2002)."

L76-80: "In June 1987, low water levels within the Grímsvötn subglacial lake due to a jökulhlaup nine months previously enabled mapping of the lake bed with RES and active seismic observations (Björnsson, 1988; Gudmundsson, 1989). Taken together with knowledge of the thickness of the overlying ice, changes

in the volume of the subglacial lake can be measured by observing changes in surface elevation (Björnsson, 1988; Gudmundsson et al., 1995).".

L80-81: This is a slightly vague statement. I suggest change to: "Mapping the lake bottom during low water levels enabled accurate quantification of lake volume change that has previously not been possible beneath ice cauldrons.".

L81:84. Suggest a rewording "However, the relationship between surface elevation within an ice cauldron and the volume of the subglacial lake beneath is not clear. Intense melting at the bed and strongly converging ice flow leads to substantial spatial and temporal variations in glacier thickness above the lake, which is particularly true after jökulhlaups when the walls of the ice cauldrons are much steeper.".

L85-101: Suggest to combine these sentences into a single paragraph. Suggested rewording is provided below.

L85-91: This section is a slight repetition of the previous paragraph and can be shortened: suggested rewording: "Despite these drawbacks, the volume of water released through jökulhlaups can be quantified by monitoring changes in the surface elevation of the ice cauldrons. For example, the surface elevation of the Skaftá cauldrons have been regularly monitored since the late 1990s using GNSS, airborne radar altimetry and additional Digital Elevation Models (DEMs) from various sources (Guðmundsson et al., 2007; Guðmundsson et al., 2018; http://jardvis.hi.is/skaftarkatlar_yfirbord_og_vatnsstada).".

L92: "In Iceland, attempts to"

L93: "RES data were"

L94: "This particular jökulhlaup destroyed"

L95-98: "Subsequently, RES data have been acquired up to twice a year over the same survey lines covering the Mýrdalsjökull cauldrons, with the aim of detecting abnormal water accumulation at the glacier bed (Magnússon et al., 2017; in review). This novel approach to monitoring subglacial lake activity has now been applied to the ESC, where RES data has been acquired annually since June 2014."

L100-101: "The unusually long pause as well as the insignificant rise in ESC surface elevation since 2011 motivated the acquisition of annual RES data."

L101: Suggested start of new paragraph which highlights the aims of this paper.

L101-103: "In this paper, the results of the annual RES surveys over the ESC are presented. Firstly, the RES data are used to derive annual DEMs of the bedrock beneath the cauldrons, which are then used to estimate the area, volume and shape of the lake every year between 2014 and 2020."

L103: "Secondly, we present a…".

L105: "2018, with a maximum discharge…"

L106-107: "This provides a unique insight into how the rapid drainage of a subglacial lake, whose geometry has been mapped using RES data, influences elevation changes at the surface of 200-400 m thick ice."

L107-110: Rather than providing the conclusions here, I suggest the final sentence of the introduction summarises the main aim of the paper: "Finally, the combination of annual lake volume estimates and surface lowering following the jökulhlaup events is used to demonstrate the applicability of repeat RES surveys as a tool for monitoring water accumulation and drainage beneath ice cauldrons in Iceland.

**Data and Method**

L113-122: Have you considered presenting this information as a table? Example below. I found it useful generating this table as I read through the subsequent data processing, analysis, and results sections. This means you could then condense this paragraph significantly.

| Survey Year | Date | Additional Details |
|---|---|---|
| 2014 | 5th June | Original RES survey lines. |
| 2015 | 3rd June | Repeat survey lines from 2015. |
| 2016 | 9th June | Large crevassing prevented some of the RES profiles from being surveyed. |
| 2017 | 7th June | Supraglacial lake formation and covering of snow over winter led to some RES profiles becoming defect. |

| 2018 | 4th June | Supraglacial lake formation and covering of snow over winter led to some RES profiles becoming defect. The density of the survey lines were doubled (200-250 m between profiles). |
|------|----------|-----------------------------------------------------------------------------------|
| 2019 | 31st May | An englacial water body probably formed tens of meters below the surface, affecting the RES measurements. |
| 2020 | 3rd June | An englacial water body probably formed tens of meters below the surface, affecting the RES measurements. |

L115-116: "This profile grid has since then been re-measured as accurately as possible every year Figs. (2-4)." Could you provide some further detail here? Did you have an automated GPS tracker? Were there significant offsets between years?

L123-125: "The RES data were acquired using standard surveying practices developed previously in Iceland (e.g. Björnsson and Pálsson, 2020; Magnússon et al., in review). The low frequency pulsed radar transmitter (5 MHz centre frequency) and receiver unit were placed on separate sledges, 35-45 m apart, in a single line and towed along the ice surface using a snowmobile." You might also want to adda sentence here stating why a 5 MHz radar was used as opposed to slightly higher frequencies.

L127-130: "The radar transmits a pulse which is then detected at the receiver. To increase the Signal-to-Noise Ratio (SNR), 256 or 512 measurements are stacked. As the system is towed along the ice surface, a 2D backscatter image is created which gives each RES measurement location on the x-axis and the travel time of the backscattered pulse on the y-axis."

L129-130 I suggest moving "but receiver measurement is triggered by the direct wave propagating along the surface from the transmitter." To the next paragraph where it is discussed further.

L129-131: I think this sentence could be made clearer. How did you measure the separation between the transmitter and receiver? If my understanding is correct, I would suggest the following change: "The centre position, **M**, between the transmitter and receiver for each RES acquisition was derived from the DGNSS positions of the snowmobile and receiver unit. By knowing half the separation between the transmit and receiver units, and the distance between the receiver and the snowmobile (~20 m), the position of **M** can be found."

L136: "(the sounding plus processing time of the stacked measurements varies by ~1 s)"

L144-145: Here I suggest incorporating part of the paragraph above: "The receiver measurement is triggered by the direct wave that propagates along the ice surface from the transmitter and is estimated as the average waveform measured with the RES over several km-long segments. This is then subsequently subtracted from the corresponding RES measurement.".

L146: Is your amplification of the signal relative to depth a simple range correction (i.e. geometrical spreading)? What is your scaling factor?

L146-149: "The 3D location of **M**, the transmitter and the receiver were used as inputs…"

L150: "assuming a radar signal propagation velocity through glacier ice ($c_{gl}$) of $1.68 \times 10^8$ m s$^{-1}$"

L152: "and a 500 m radar beamwidth illuminating the glacier bed."

L153-156: "The x-axis corresponds to the profile length with a horizontal resolution of 5 m, and the y-axis corresponds to m a.s.l. with a vertical resolution of 1 m. This corresponds roughly to the horizontal sampling density when measuring with a ~1 s pulse interval at ~20 km hour$^{-1}$, and an 80 MHz vertical sampling rate (in 2014-2017; it is 120 MHz for a new receiver unit used in 2018-2020)". Is the new receiver unit used in 2018-2020 a different model to that used in the previous campaigns?

L164-166: Without being able to see the Magnússon et al. (in review) paper I do not know how these steps were conducted. It would be best to briefly expand on these here.

L167-193: I think Section 2.2 can be condensed and made clearer – I have made some suggestions below.

L167: When you say the profiles were projected onto the same length axis, do you mean truncated so that the profiles can be compared directly i.e. they only represent overlapping areas? Unclear as written.

L167-176: Suggested change: "Each RES profile containing the traced reflections from the subglacial lake and bedrock are projected onto a common profile, where profiles 2014-2017 are projected onto the 2014 profile and profiles 2018-2020 projected onto the 2018 profile. When comparing the traced bedrock reflections (i.e. outside the rim of the ESC) between surveys, the median elevation difference was <2.5 m

for profiles 2015-2020 relative to 2014 (in 2018 and later the shift is obtained from comparison with an interpolated bedrock DEM based on surveys from previous years). Assuming these bedrock areas are unchanged between surveys, we correct each survey 2015-2020 by this vertical bias."

L180-181: How do you approximate the lake area in between RES profiles? Have you manually drawn the boundary (this could be subjective)? Whilst you discuss this for individual years in the sentences below, it would be useful to know what general procedure you adopted. An indication of uncertainty (even just a crude approximation) would also be useful.

L183: "guide the"

L184: "lake margin"

L185: Remove "where this limitation applied to the 2014 survey"

L185-186: "was however guided by the RES data alone."

L187 and L189: "Corrupted" isn't necessarily the right word here, I would use "obstructed".

L189: "is expected to be more accurate than the preceding year."

L191: "somewhat uncertain" is slightly vague, are you able to quantify how less accurate it is?

L193: Do you mean the upper limit of its expected size or the upper limit of the expected accuracy?

L197: "within the ESC and below the subglacial lake"

L197-198: "The traced bedrock reflections has good coverage across the bedrock beneath…"

L198-199: "In addition, the bedrock elevation beneath the cauldrons has been measured… "

L200: remove "fortunately"

L200: Did you validate your interpolation using the borehole measurements? If not, this could be a useful piece of analysis to validate your bedrock map and provide an indication of bedrock DEM uncertainty.

L201-202: "has been constructed using the kriging interpolation method (processed using Surfer 13 © Golden Software LLC)"

L203-205: Combine "The filtered…at that time" with paragraph below.

L204-206: I would remove "An independent…(see section 4.1.3)" as it is not relevant here.

L216-217: "…were then differenced from the interpolated bedrock DEM to obtain…"

L217-219: "The lake outlines are converted to points and are prescribed a lake thickness of zero before interpolating each lake thickness map…"

L227: Change subheading to "Elevation changes and released volume of water juring jökulhlaups in 2015 and 2018"

L229: Suggested change to "acquired by the TanDEM-x and TerraSAR-X spaceborne bistatic interferometer"

L231: "Differencing the two DEMs reveals…"

L235-236: Remove "a correction was deployed"

L236: "a ~500"

L238-239: "reference area and then subtracted from the elevation differences between the two DEMs."

L241: "4 June during"

L259-264: Does the lake margin correspond to the area of elevation change in 2015? If so, you could use this to constrain the elevation change area in 2018.

L265-266: Where do these biases come from?

L267269: Please state exactly why this correction needs to be applied.

L259-275: This feels like it should be in the results section. I suggest putting into the results under the heading "Water volume released during jökulhlaup in 2015 and 2018".

***Results***

L279: "At the time of this observed maximum lake area in 2015,…"

L280-281: "In comparison, the lake had expanded to 3.2 km$^2$ in June 2018, two months priors to the 2018 jökulhlaup."

L281-283: Suggested change to: "The strong positive linear correlation between the area and volume of the subglacial lake is demonstrated in Fig. 6i.". This may also be impacted by the fact the same bedrock DEM is used throughout. Could the bedrock have changed over the acquisition period from e.g. subglacial volcanic activity?

L285: "lake volumes"

L290-291: Remove "indicating the applicability of our RES survey approach to evaluate the expected hazard from a jokulhaup."

L292-314: This paragraph is interesting. You explain the offset of the 2014 and 2015 lake volumes extracted from the RES surveys by the different measurement densities, but 2016 and 2017 were also coarsely sampled. Could it be then that the offset is simply due to slightly different subglacial lake refilling rates? A better comparison would therefore be the linearly regress 2015-2017 and 2018-2020 separately. You could even try the same regression with the 2014-2015 data to see the result of this as well.

L321: "The shape of the subglacial lake margin also differed between 2015 and 2018. "

L321-323: "Steep side walls surrounded the bulk of the lake, although the thickness of the lake was typically 10-30 m away from the lake margins (see Fig. 6f and left side of Fig. 7b)."

L324-325: You should also reference Fig. 8a-b as this is where we see the greater lake thickness in 2018 – Fig.9a only shows the volume calculations.

L328-329: "The outward migration of the lake margin, typically by 50-150 m, was characterised by the outward propagation of the steep sided ice walls that defined the lake margin.".

L328-332: I would caution this discussion of the steep-sided subglacial lake walls. The interpolation was guided by setting the rim of the lake walls to zero. Whilst there is clear good evidence for steep-sided ice walls, there may also be interpolation error that could bias some of the lake shapes.

L353-355: It appears to me that the maximum surface lowering regions are at the centre where the lake is not at its thickest – the largest lake thickness is to the eastern side of the ESC.

L355-369: This paragraph is a useful summary, but I think you the need make it clear what the terms 'thickening' and 'thinning' are referring to. If, as is stated in text, this refers to changes in ice thickness, you should also then state that this assumes the lake has completely drained. This means that for a completely drained lake, if the difference between the lake thickness and the surface lowering is positive, then the ice has thickened. I would try to reword this paragraph to make this more clear. Do you have additional data to suggest that the entire lake has drained?

*Discussion*

L371-383: This paragraph provides results that are then discussed in the subsequent sections. To improve the flow of the text, it might better to have this as a separate section (4.1.5.) and use it to summarise the contributing errors and state the accuracy of 10-20%. The authors may leave this suggestion if they do not see it as useful.

L374-379: This is a long sentence. Suggest shortening: "The RES surveys before the jökulhlaups in 2015 and 2018 show a good agreement with the derived surface lowering patterns (Fig. 8–9). Together with a close linear relationship between the time elapsed since the previous jökulhlaup (Fig. 9b) except when the lake is small and not hazardous suggests lake volume errors from RES measurements are typically 10-20%."

L394: Has the 20% error been calculated based on the data gaps created from the supraglacial lakes? You should state this here to be clear.

L393-399: Even though this discussion refers to RES data gaps, I think it is best placed under Section 4.1.4 and I would suggest moving it there.

L401-404: Suggested shortening to: "In most glaciological applications, only 2D migration of RES data is possible but even this requires the assumption that all radar reflections originate from directly beneath the survey profile. This is often not the case beneath glaciers that flow over volcano's, where the subglacial topography is particularly complex."

L405: Remove "but much smaller when the slope and profile directions are in parallel" and combine with sentence spanning Lines 405-406.

L406-410: "If the traced reflective surface is not directly beneath the RES profile but cross-track, the obtained ice thickness is underestimated and the mapped surface below the profile is estimated to be too high."

L410: "This has been shown using an experiment…"

L411-412: "Similar results were obtained in a recent study on Mýrdalsjökll ice cap (S-Iceland) which has a similar topographic setting to the ESC (Magnússon et al., in review). In that study, 2D migrated profiles were found to be 10 m higher than the bedrock DEM obtained from 3D migrated data."

L418: "allow for safe acquisition of data for 3D migration without reasonable effort"

L419-421. The two effects probably mostly cancel each other out, but given the complexity of the subglacial topography, I'm not sure this can be stated with confidence. I would instead note that in general the two effects cancel out, but across the complex topography described in this study, the effect is likely to be similar to the Magnússon et al. (in review) study.

L423-428: I think this paragraph should be placed in Section 4.1.1. as it discusses the effects of data gaps on underestimating subglacial topographic peaks. Lines 424-425 can be retained in this section as it is pertinent to the effects of the 2D migration processing.

L432: Remove "the"

L433: "that the actual"

L440-441: "shift both the lake roof and the bedrock in the same…"

L443-446: I think this section needs rewording. Suggestion: "If $c_{gl}$ is too large, lake thickness is overestimated, and it is underestimated if $c_{gl}$ is too low. For example, if the true value of $c_{gl}$ is $1.60 \times 10^8$ m s$^{-1}$ but a value of $1.68 \times 10^8$ m s$^{-1}$ is used, the lake volume would be overestimated by ~5%. Considering the upper limit of $c_{gl}$ is $1.70 \times 10^8$ m s$^{-1}$, a significant overestimate of $c_{gl}$ is unlikely."

L451: "…value of $c_{gl}$ may differ significantly between some lake roof and bedrock measurements, leading to larger uncertainties at locations where such water bodies exist."

L452-454: Are crevasses persistent around the cauldron rim or are they transient features? Are they seasonally filled in with snow?

L468: "hilly topography" is a little colloquial, I suggest changing to "undulating lake surface topography"

L475: Remove "minority" to "small number".

L485-492: This is a worthwhile discussion but should be moved, possibly to Section 4.3.

L500-504: I don't think this part is necessary, I would focus on the results you have presented in the paper and use these to develop your discussion.

L505-507: Reword to: "Temperature profiles within the subglacial lakes beneath the Skaftá cauldrons have revealed temperatures of 3-5°C that are mostly independent of lake depth, thus enabling effective convection to take place (Jóhannesson et al., 2007; unpublished data at the IMO). Chemical…".

L509: Remove "a".

L518-519: Here, you suggest enhanced geothermal activity is likely the reason for enhanced basal melting. Do you have additional data/publications to back this up? Could other factors play a role?

L537: Remove "of the tremor"

L558-359: "The GNSS station based on the ESC surface, operated by IMO, was operating during both jökulhlaups.".

L539-541: I suggest remove this sentence as it is redundant.

L547-549: "The differences in lake area between 2015 and 2018 can explain the difference in surface uplift rates but not the slower initial subsidence in 2015".

L550-561: This paragraph could be combined with the preceding paragraph and shortened.

L564-568: "Such a deceleration is not observed in 2015 and may be caused by floating ice atop the 10-20 m thick water layer moving against the bedrock a few hundred metres south of the GNSS station. Whilst a supraglacial lake inhibited complete mapping south of the GNSS station, traced reflections from RES data 450 m south of the station suggest the ice was grounded at this location.".

L574: I'm not sure I see where the 2018 subsidence sped up again. Is it just before the 4-day mark? Either way, it seems relatively insignificant – the fact it is not apparent in the 2018 data should be highlighted as a difference between 2015 and 2018.

L585: Remove "near"

L593: "cauldron centre to be substantially…"

L595: "of ice at a given"

L596-598: The thickening is also partially due to the convergence of ice flow into the cauldron and should acknowledged here.

L607: "motion decreases from"

L616-631: I think this is an interesting place to end, but I would also add that the RES survey design could also be improved so that the subglacial lake can be mapped at sufficient resolution to remove interpolation errors.

**Conclusions**

L633-634: "Repeat RES surveys over the Eastern Skaftá Cauldron (ESC) and a comparison with surface lowering during jökulhlaups was used to measure the volume of a subglacial lake beneath the ESC every year between 2014 and 2020. This novel data set has been used to demonstrate the applicability of repeat RES surveys for quantitative monitoring of subglacial lake volumes."

L632-646: This is more of a summary than a conclusion. I think some key information is missing. For example, what are the uncertainties of the RES data, by how much has the lake volume changed over time and information relating to the two jökulhlaups (2015 and 2018). You should then frame these key results into a brief summary of the advantages and limitations of the repeat-RES approach and suggest possible future developments of the technique.

**References**

L766: Change "USAGE" to "ISAGE"

**Figures**

Figure 1: Good figure overall. Could you label the red boxes in panel (a) to make it clear which each is referring to (i.e. red box on inset panel is referring to panel (a) and inset in panel (a) is referring to panels (b) and (c))? Could you label ESC and WSC instead of "Skaftá cauldrons"? Is the glacier outline from GLIMS and does it need a reference? For panels (d) and (e), it is best to have the dates on the images to avoid excessive reference to the figure caption, with exact dates if you have them. For demonstrating the viewing angle, I think you should mark at the apex of the red line that this is the position of the camera, with an arrow indicating viewing direction.

Figure 2: Interesting figure. The red box on the inset map of Vatnajökull is annotated "ESK" which I assume should be "ESC". In panel (a), I worry that red and green lines cannot always been seen by those with colour blindness, possibly change to blue or black? I would change the map of Vatnajökull to panel (a) and have the map of the profile grid as panel (b). Thus, change "(located on corner inlet)" to "(location in **a**). The legend details are a bit scattered. It might be better to put all of these (e.g. bed reflection, surfaces) at the bottom of each panel to avoid cluttering the graphs.

Figure 3: Good figure

Figure 4: Overall a useful figure. I wonder if panel **h** would be more instructive if it showed the survey lines in the same colour e.g. black, or possibly showing the sparse (2014-2018) and dense (2018-2020) survey lines in different colours.

Figure 5: Good figure

Figure 6: Good figure overall. Scale bar needs a label "Lake Thickness (m)" and the bedrock DEM also needs a corresponding color scale. In the figure caption, change "ESK" to "ESC". Red might not be the best colour to use for the survey lines, suggest change to dotted black (or another suitable colour).

Figure 7: Very interesting figure. My only concern is that the figure suggests the surface lowering occurs without a change in the lake water level. You could caution this on the figure by writing the date the lake volume has been estimated.

Figure 8: Each color scale should have a label. The red and cyan lines could also be annotated on the map rather than having to refer to the figure caption. Otherwise, a good overview of the surface changes observed.

Figure 9: I have no major problems with this figure. If possible, it would be best to move the legend above the figure to avoid overfilling panel (a).

Figure 10: Was the GNSS repositioned to the exact same location before each jökulhlaup event? I would also state what the symbols mean in the figure caption or legend to make it easier for the reader to understand.

---

## Author Comment (AC1)

**Response to review by Anonymous Referee #1 of manuscript TC-2021-65:**

**Development of a subglacial lake monitored with radio echo sounding: Case study from the Eastern Skaftá Cauldron in the Vatnajökull ice cap, Iceland**

**General Comments**

*This paper presents a new technique to monitor subglacial lake activity in Iceland using repeat Radio Echo Sounding (RES) surveys. The authors demonstrate the applicability of the new technique by deriving lake area and volume estimates between 2014 and 2020 for the subglacial lake below the Eastern Skaftá Cauldron (ESC), Vatnajökull. They then derive surface elevation changes from InSAR and airborne radar altimetry and compare these to the lake volume estimates to characterise two jökulhlaups which occurred in 2015 and 2018. Detailed long-term measurements of subglacial lake activity are rare but vital in understanding hazards posed by glacio-volcanic activity (e.g. jökulhlaups). The underpinning methods appear robust with some clarification required in placed and I've detailed these in the Technical Corrections below. In this context, the study is unique and worthy of publication in The Cryosphere, but subject to some revisions detailed below.*

Thanks for these positive remarks.

**Specific Comments**

*The paper as it stands is quite long, stretching over 30 pages, and can be shortened – I've tried to provide suggestions in my Technical Corrections below to help the authors achieve this. For example, the Discussion section is 8 pages long and can be condensed by writing in a more concise manner. In particular, the authors sometimes discuss topics that are not relevant to a particular section and thus break up the flow of the discussion. For example, the discussion of uncertainties in the RES data pertaining to supra- and englacial water bodies is followed by a discussion on glacier hydrology and how this impacts jökulhlaup processes. These items should be discussed separately. Other suggested changes to the text are: breaking up the Results section into sub-sections and moving the discussion of RES errors into the methods section. In general, by ensuring each section is more focussed on developing the discussion pertaining to a particular section, each section will likely reduce in size and lead to a more concise paper overall.*

In general we have followed most of the suggestions by reviewer to shorten the text were possible. In some cases the suggested replaced text was simply too short to describe the intended message, and therefore required a few extra words. The reviewer has also pointed at several locations where he

thinks additional information are needed. In most of these we agree with the reviewer and have added the requested information. We also added a table requested by reviewer.

We generally follow the instruction regarding reorganisation of the text, including addition of sub-sections in the results chapter. The comments by reviewer regarding the uncertainties made us realise that this part of the old manuscript was quite confusing, making the reviewer misunderstand our approach regarding the uncertainties. In our study we do not try to estimate the uncertainties on the lake volume derived from the RES by estimating and combining various uncertainty parameters likely to contribute to errors in our RES survey approach. As explained in the revised manuscript we consider it difficult to get a meaningful uncertainty estimate with such an approach, because it is hard to quantify the different factors that can contribute to this uncertainty and the dependency between different uncertainty factors is unclear, which further complicates calculation of a single uncertainty value for the derived lake volumes. Instead, the lake volumes derived from the RES data were validated by comparing them with the volume of water released during jökulhlaups, obtained from measured surface lowering. Clarifying this has resulted in a short additional section in the method chapter as well as extension of the results chapter to include the results of the validation. In section 4.1 (and subsections) on the limitation of our survey approach we now avoid talking about uncertainties but instead talk about likely errors to avoid the confusion that we trying to derive the uncertainty of our approach. We do however end section 4.1 with short discussion on how the survey limitations relate to the results of the validation.

Due to all this both methods, results and conclusions (discussed further below) have been slightly extended, while this length of both the introduction and discussion sections has been slightly shortened. In total the manuscript text has been extended by 5 lines and with the new table the submitted manuscript extends few lines into the 31st page but was 30 pages in last submission. Note also that due some text cutting the figure, which used to be nr. 7 is initially referenced later in the text and is now therefore nr. 9. Consequently, the figures which used to be nr. 8 and 9 are now number 7 and 8.

*Further, the writing in places is vague. The authors often describe the results with words such as 'seemingly' and 'almost' without providing quantitative statements from the results to back up their claims. In general, this can be resolved with a few simple calculations, but in other places the authors should consider referencing other suitable data sets or studies. This is particularly true in the Conclusion section which is written as more of a summary. Here, it would be better to provide some of the headline results from the study, backed up by quantitative statements, and state what the key take-home messages are from this study. I think the two main conclusions are:*

> • *A new method has been developed for long-term monitoring of a subglacial lake which helps improve understanding of jökulhlaups.*
> • *A unique analysis of jökulhlaups in 2015 and 2018 has been conducted that uses estimates of lake volume and surface elevation changes to understand the lake development before the jökulhlaups.*

We try to avoid the kind of wording mentioned above in our revised manuscript. The conclusion has been slightly extended even though we prefer not to repeat the numeric results given in both the abstract and result sections, apart from the results of the validation supporting our conclusions in relation to our RES survey approach. The conclusions of the revised manuscript include a take-home message regarding the applicability of our approach for monitoring the lake beneath ESC, what we consider the limitation of the method as well as pointing how our approach could be improved to make it more applicable elsewhere, particularly for lake beneath smaller ice cauldron. The take home-massages of this study are, however, not only related to possible future developments of the technique. For those more interested in the nature of subglacial lake beneath ice cauldrons, we also point out how this data set provides a completely unique view on such lake and what new study opportunities the presented data opens regarding how these lakes behave both in between and during jökulhlaups.

*The methods underpinning this study are generally robust. However, the estimation of lake volume changes is dependent on having an accurate bedrock DEM. Is it possible that the bedrock outside and below the lake has changed over the course of the study period due to subglacial volcanic activity? One way in which you could quantify this is to find overlapping RES measurements representing the bedrock beneath the lake and calculate the vertical deviation between surveys. You have done this for areas outside the lake, but it might be possible to look at this below the lake.*

Our methodology is based on detecting the water as deviations in the elevation of reflective surface from the temporal minimum corresponding to bedrock surface (the correction you are pointing at refers to areas we are sure that were never covered by the lake during our study). This approach can therefore not discriminate between new volcanic formations and the lake and we would not obtain reflections from volcanic formation within the area covered by the lake at all time. However, there are no evidences of significant eruptions taking place during our study period. The only phenomena observed during the study period that is similar to what is seen in volcanic eruptions are powerful low frequency seismic tremor pulses that have often been observed near the end of jökulhlaups, commonly related to boiling in the geothermal system triggered by pressure relief (Eibl et al., 2020;

Guðmundsson et al., 2013b). The alternative explanation that this may be tremor from minor eruptions has been suggested (Guðmundsson et al., 2013b). However, if these tremor pulses were related to volcanic eruptions, they were very minor and short-lived, producing a net volume probably well within 1 million m$^3$ (which may even to large extend be flushed out by the jökulhlaup) and would therefore be insignificant in comparison to all water volumes presented here, as well as in terms of the energy released by the geothermal system. The fact that this hypothesis (which is not our own) has not been properly published in a peer reviewed paper along with the already extensive length of the paper are the main reasons we prefer not discussing it in our paper."

*Overall, if the authors can incorporate these suggestions and those in the Technical Corrections below, I'm sure this will make a very nice paper.*

**Technical Corrections (References to line numbers in preprint)**
**Abstract**
*L13: "The ESC is a ~3 km"* Done.

*L14: "subglacial lake"* Done.

*L15: "summer"* Done.

**Introduction**
*L26-27: "Subglacial lakes have been directly and indirectly observed beneath both temperate and coldbased glaciers. The sudden release of water from such lakes can lead to floods, commonly referred to as jökulhlaups, and can be of variable magnitude.". Reference?*

Replacement done but we don't think we need a reference to state that jökulhlaups are variable in size.

*L27-28: "In warm-bedded glaciers, jökulhlaups are known to enhance basal sliding and increase glacier flow over a period of days (e.g. Einarsson et al., 2016)"*

Done.

*L32: I suggest beginning this paragraph with a short sentence summarising how subglacial lakes have previously been observed before launching into the paragraph, e.g. "Documenting the existence of subglacial lakes has been achieved using a combination of radio-echo sounding (RES) data and satellite remote sensing, but routine monitoring of such lakes remains a difficult task.".*

Done.

*L34: The exact quotation here should be "thick water layer beneath the ice".*

We apologise for this sloppy mistake. Thanks for spotting this. That is quite impressive!

*L34: "Since then,"* Done.

*L35: "RES data"* Done.

*L35-39: Suggest combining these sentences together: "However, many subglacial lakes actively drain and fill and as a result are difficult to distinguish in RES data (Carter et al., 2007; Siegert et al., 2014), hence SAR interferometry and repeat altimeter surveys have been used to identify hundreds of areas of surface elevation changes associated with active subglacial lakes in Antarctica (e.g. Gray et al., 2005; Smith et al., 2009)."*

Done. To clarify that we are not identifying the lake beneath ESC in same way as usually from RES we added in section 2.2 a small sentence stating that the roof of ESC is not revealed by a flat reflection.

*L40-47: Suggest combine with paragraph below and shorten. See next few comments.*

*L40-44: Suggested start of new paragraph: "In Iceland, subglacial lake drainage events that lead to jökulhlaups have been documented since the early 1900s (Thorarinsson and Sigurðsson, 1947; Thorarinsson, 1957) and are well known to cause widespread destruction of farms and infrastructure, as well as loss of life."*

Done, except we are not certain about casualties in relation to jökulhaups from subglacial lakes, hence we prefer stating the threat to lives rather than the actual loss of lives.

*L44-47: "Subglacial lakes in Iceland are formed through localized geothermal activity, where enhanced basal melting forms topographical depressions on the glacier surface (ice cauldrons), creating a low in the hydrostatic potential and promotes water accumulation from both the glacier surface and bed (Björnsson, 1988).*

The suggested text inserted, but one sentence later than the text it replaced.

*L49: "the two skaftá cauldrons, denoted as the Eastern Skaftá Cauldron (ESC) and the Western Skaftá Cauldron (WSC).".*

Done.

*L50: Change "within" to "below".*

We use the word within, since it was not certain, whether it was subglacial or not.

*L50: "and has historically been the source of large jökulhlaups that have drained from beneath the Skeiðarárjökull outlet glacier in Southern Vatnajökull (Wadell 1920)."*

The suggested text omits the message, which we want to include here, that people knew about Grímsvötn due to the jökulhaups before they were directly discovered by Wadell and Ygberg in 1919. We therefore stick with our original text.

*L53-56: Suggest combining these sentences together: "The first direct observation of the ESC was from a photograph taken in 1938 whilst the WSC was first observed in 1960 (Guðmundsson et al., 2018), suggesting the ESC and WSC have not always co-existed.".*

We rather prefer explaining how we know they have not always co-exist (from the AMS photographs in 1945-46, the 1938 photograph does not proof this since it does not show the area of WSC) as well including info on the development of ESC. We however included some of the suggested text-cuts.

*L56-61: "The geothermal power beneath Grímsvötn has been estimated from the volume of water discharged through jökulhlaups and surface mass balance and is estimated to be approximately 1500-2000 MW (Björnsson, 1988; Björnsson and Guðmundsson, 1993; Guðmundsson et al., 2018; Reynolds et al., 2018; Jóhannesson et al., 2020). This is likely to be similar below both the ESC and WSC making the region some of the most powerful geothermal areas in Iceland.".*

Changed according to the above suggestion with some minor further changes.

*L62: "which resulted in 4.7 km$_3$ and 3.4 km$_3$ of water being released, respectively (Gudmundsson et al., 1995; Björnsson, 2002)."*

Done.

*L76-80: "In June 1987, low water levels within the Grímsvötn subglacial lake due to a jökulhlaup nine months previously enabled mapping of the lake bed with RES and active seismic observations (Björnsson, 1988; Gudmundsson, 1989). Taken together with knowledge of the thickness of the overlying ice, changes in the volume of the subglacial lake can be measured by observing changes in surface elevation (Björnsson, 1988; Gudmundsson et al., 1995).".*

Change approved apart from the final part of the sentence.

*L80-81: This is a slightly vague statement. I suggest change to: "Mapping the lake bottom during low water levels enabled accurate quantification of lake volume change that has previously not been possible beneath ice cauldrons.".*

It seems that the statement was so vague that it was misunderstood, given the suggested replacement, which does not include our intended message. We therefore simply omitted it.

*L81:84. Suggest a rewording "However, the relationship between surface elevation within an ice cauldron and the volume of the subglacial lake beneath is not clear. Intense melting at the bed and strongly converging ice flow leads to substantial spatial and temporal variations in glacier thickness above the lake, which is particularly true after jökulhlaups when the walls of the ice cauldrons are much steeper.".*

Apart from the final part of the sentence this was mostly according to suggestion.

*L85-101: Suggest to combine these sentences into a single paragraph. Suggested rewording is provided below.*

We prefer keeping the discussion on surface changes in ice cauldrons in single paragraph, hence the part which used to be in line 85—91 is now a part of the paragraph, which used to precede it.

*L85-91: This section is a slight repetition of the previous paragraph and can be shortened: suggested rewording: "Despite these drawbacks, the volume of water released through jökulhlaups can be quantified by monitoring changes in the surface elevation of the ice cauldrons. For example, the surface elevation of the Skaftá cauldrons have been regularly monitored since the late 1990s using GNSS, airborne radar altimetry and additional Digital Elevation Models (DEMs) from various sources*

*(Guðmundsson et al., 2007; Guðmundsson et al., 2018; http://jardvis.hi.is/skaftarkatlar_yfirbord_og_vatnsstada).".*

Changed accordingly apart from minor further modification.

*L92: "In Iceland, attempts to"* Done.

*L93: "RES data were"* Done.

*L94: "This particular jökulhlaup destroyed"* Done.

*L95-98: "Subsequently, RES data have been acquired up to twice a year over the same survey lines covering the Mýrdalsjökull cauldrons, with the aim of detecting abnormal water accumulation at the glacier bed (Magnússon et al., 2017; in review). This novel approach to monitoring subglacial lake activity has now been applied to the ESC, where RES data has been acquired annually since June 2014."*
Done.

*L100-101: "The unusually long pause as well as the insignificant rise in ESC surface elevation since 2011 motivated the acquisition of annual RES data."*
Done.

*L101: Suggested start of new paragraph which highlights the aims of this paper.*
The last part with the highlights is now a separate paragraph.

*L101-103: "In this paper, the results of the annual RES surveys over the ESC are presented. Firstly, the RES data are used to derive annual DEMs of the bedrock beneath the cauldrons, which are then used to estimate the area, volume and shape of the lake every year between 2014 and 2020."*
The first part changed according to suggestion. We are however assuming that the bedrock remains fixed during the study period, hence referring annual DEMs of bedrock does not accurately describe the presented result, and therefore the latter part resembles more our previous text.

*L103: "Secondly, we present a…".* Done.

*L105: "2018, with a maximum discharge…"* Done.

*L106-107: "This provides a unique insight into how the rapid drainage of a subglacial lake, whose geometry has been mapped using RES data, influences elevation changes at the surface of 200-400 m thick ice."*
Replaced with suggested text including minor changes.

*L107-110: Rather than providing the conclusions here, I suggest the final sentence of the introduction summarises the main aim of the paper: "Finally, the combination of annual lake volume estimates and surface lowering following the jökulhlaup events is used to demonstrate the applicability of repeat RES surveys as a tool for monitoring water accumulation and drainage beneath ice cauldrons in Iceland.*
Changed to a similar manner as suggested, but with replaced text more precise than suggested. Due to some limitations of our methods it is not applicable for all ice cauldrons in Iceland, hence we prefer not such general statement as the last words in the suggested text.

**Data and Method**

*L113-122: Have you considered presenting this information as a table? Example below. I found it useful generating this table as I read through the subsequent data processing, analysis, and results sections. This means you could then condense this paragraph significantly.*

**Survey Year Date Additional Details**

*2014 5$_{th}$ June Original RES survey lines.*

*2015 3$_{rd}$ June Repeat survey lines from 2015.*

*2016 9$_{th}$ June Large crevassing prevented some of the RES profiles from being surveyed.*

*2017 7$_{th}$ June Supraglacial lake formation and covering of snow over winter led to some RES profiles becoming defect.*

*2018 4$_{th}$ June Supraglacial lake formation and covering of snow over winter led to some RES profiles becoming defect. The density of the survey lines were doubled (200-250 m between profiles).*

*2019 31$_{st}$ May An englacial water body probably formed tens of meters below the surface, affecting the RES measurements.*

*2020 3$_{rd}$ June An englacial water body probably formed tens of meters below the surface, affecting the RES measurements.*

Done including minor modifications on the table text.

*L115-116: "This profile grid has since then been re-measured as accurately as possible every year Figs. (2-4)." Could you provide some further detail here? Did you have an automated GPS tracker? Were there significant offsets between years?*

Further info on this is now included.

*L123-125: "The RES data were acquired using standard surveying practices developed previously in Iceland (e.g. Björnsson and Pálsson, 2020; Magnússon et al., in review). The low frequency pulsed radar transmitter (5 MHz centre frequency) and receiver unit were placed on separate sledges, 35-45 m apart, in a single line and towed along the ice surface using a snowmobile." You might also want to add a sentence here stating why a 5 MHz radar was used as opposed to slightly higher frequencies.*

Suggested text adopted in addition to justification on why 5HZ were used.

*L127-130: "The radar transmits a pulse which is then detected at the receiver. To increase the Signal-to-Noise Ratio (SNR), 256 or 512 measurements are stacked. As the system is towed along the ice surface, a 2D backscatter image is created which gives each RES measurement location on the x-axis and the travel time of the backscattered pulse on the y-axis."*

*L129-130 I suggest moving "but receiver measurement is triggered by the direct wave propagating along the surface from the transmitter." To the next paragraph where it is discussed further.*

*L129-131: I think this sentence could be made clearer. How did you measure the separation between the transmitter and receiver? If my understanding is correct, I would suggest the following change: "The centre position, **M**, between the transmitter and receiver for each RES acquisition was derived from the DGNSS positions of the snowmobile and receiver unit. By knowing half the separation between the transmit and receive units, and the distance between the receiver and the snowmobile (~20 m), the position of **M** can be found."*

*L136: "(the sounding plus processing time of the stacked measurements varies by ~1 s)"*

*L144-145: Here I suggest incorporating part of the paragraph above: "The receiver measurement is triggered by the direct wave that propagates along the ice surface from the transmitter and is estimated as the average waveform measured with the RES over several km-long segments. This is then subsequently subtracted from the corresponding RES measurement.".*

The text corresponding to lines 123-145 (which last 6 comments refer to) in previous version has been rewritten for clarification, since it seems from some of the comments above related to it, that it was not all correctly understood. The new text is partly based on suggested replacement were it fits the intended meaning, but otherwise further clarifications are provided including clarification on how the antenna separation (referred to as *a* in the current text) and distance from receiver to snowmobile (referred to as *b* in the current text) are derived.

*L146: Is your amplification of the signal relative to depth a simple range correction (i.e. geometrical spreading)? What is your scaling factor?*

The amplification is done by plotting the average absolute signal strength for several km long segments as a function of travel time. This generally reveals approximately a linear relation on a log scale graph up to certain travel time where all backscatter is overprinted by noise and the absolute signal strength becomes approximately constant with travel time. For travel time smaller than this noise level travel time we estimate from this plot and apply a linear (on log scale) gain function of the travel time and for larger travel times constant gain is applied. We rather prefer skipping this detail in already too long manuscript since this step is not very crucial for this work. The signals are generally quite clear without this amplification and this has insignificant effects on the positions of reflection traced from the 2D migrated RES-data. The main advantage of this is that it makes the 2D migrated RES profiles (e.g. Fig. 2) nicer to look at when shallow reflections are not by far brighter than the deeper ones. Applying correction using geometrical spreading, which also would effectively result in linear gain function on log scale graph, has the disadvantage that the "deep" noise will become unnecessarily bright unless a threshold travel time was defined.

*L146-149: "The 3D location of **M**, the transmitter and the receiver were used as inputs…"*

This sentence has been slightly rewritten for further clarification with reference to the distance *a*/2 to better describe how the input positions of transmitter and receiver are derived.

*L150: "assuming a radar signal propagation velocity through glacier ice ($c_{gl}$) of $1.68 \times 10^8$ m s$^{-1}$"*

Done.

*L152: "and a 500 m radar beamwidth illuminating the glacier bed."*

Done.

*L153-156: "The x-axis corresponds to the profile length with a horizontal resolution of 5 m, and the y axis corresponds to m a.s.l. with a vertical resolution of 1 m. This corresponds roughly to the horizontal sampling density when measuring with a ~1 s pulse interval at ~20 km hour$_{-1}$, and an 80 MHz vertical sampling rate (in 2014-2017; it is 120 MHz for a new receiver unit used in 2018-2020)". Is the new receiver unit used in 2018-2020 a different model to that used in the previous campaigns?*

Changed according to suggestion. We changed the word "new" to "upgraded" since this is still a receiver unit from Blue System Integration Ltd.

*L164-166: Without being able to see the Magnússon et al. (in review) paper I do not know how these steps were conducted. It would be best to briefly expand on these here.*

This article has now been accepted and should be available before this paper will published in a final format. We therefore want to avoid extending this paper by repeating this description. However in this paper the following text will be found: "Backscatter from the glacier bed is usually recognised as the strongest continuous reflections at depth in the 2D migrated amplitude images. They were traced with an automatic tracing algorithm, programmed in Matlab (®Mathworks). The algorithm traces the bed reflection from a chosen clear point of bed reflection, by using the maximum correlation with the bed reflection at the chosen starting point. The obtained traces were manually revised to reject derived traces where the algorithm failed. This process was repeated until all clear bed reflections had been traced for each profile of individual survey. At sharp turns in the survey profiles reflections were rejected. The assumption of fixed distance between transmitter and receiver fails at these turns and the 2D migration is not expected to result in an accurate depth of reflector." And later one regarding the filtering: "The traced reflections of the 2D migrated data were filtered with a 25 m wide triangular filter and down-sampled to values at 20 m interval along the profile"

*L167-193: I think Section 2.2 can be condensed and made clearer – I have made some suggestions below.*

*L167: When you say the profiles were projected onto the same length axis, do you mean truncated so that the profiles can be compared directly i.e. they only represent overlapping areas? Unclear as written.*

Further explanation has been added to clarify.

*L167-176: Suggested change: "Each RES profile containing the traced reflections from the subglacial lake and bedrock are projected onto a common profile, where profiles 2014-2017 are projected onto the 2014 profile and profiles 2018-2020 projected onto the 2018 profile. When comparing the traced*

*bedrock reflections (i.e. outside the rim of the ESC) between surveys, the median elevation difference was <2.5 m for profiles 2015-2020 relative to 2014 (in 2018 and later the shift is obtained from comparison with an interpolated bedrock DEM based on surveys from previous years). Assuming these bedrock areas are unchanged between surveys, we correct each survey 2015-2020 by this vertical bias."*

We are not sure if intension of this text was to shorten or clarify the previous one, but we think the suggested one is oversimplified in such way that it is slightly inaccurate or misleading. Hence we stick with the previous text, including some further clarifications, particularly for the first part.

*L180-181: How do you approximate the lake area in between RES profiles? Have you manually drawn the boundary (this could be subjective)? Whilst you discuss this for individual years in the sentences below, it would be useful to know what general procedure you adopted. An indication of uncertainty (even just a crude approximation) would also be useful.*

Some clarifications on this have been added. A text regarding uncertainties in lake margin area has been added at the end of the paragraph.

*L183: "guide the"*

Done.

*L184: "lake margin"*

Done.

*L185: Remove "where this limitation applied to the 2014 survey"*

Done.

*L185-186: "was however guided by the RES data alone."*

This sentence has been rephrased.

*L187 and L189: "Corrupted" isn't necessarily the right word here, I would use "obstructed".*

Done

*L189: "is expected to be more accurate than the preceding year."*

This part of the sentence has been removed because of extra sentence regarding area uncertainties at the end of the paragraph.

*L191: "somewhat uncertain" is slightly vague, are you able to quantify how less accurate it is?*

This part of the sentence has also been removed because of extra sentence regarding area uncertainties at the end of the paragraph.

*L193: Do you mean the upper limit of its expected size or the upper limit of the expected accuracy?*

Sentence rephrased to clarify.

*L197: "within the ESC and below the subglacial lake"*

Not clear what this should replace, but we have now rephrased this line for further clarification, since it seems to have caused misunderstanding.

*L197-198: "The traced bedrock reflections has good coverage across the bedrock beneath…"*

Done.

*L198-199: "In addition, the bedrock elevation beneath the cauldrons has been measured… "*

Done.

*L200: remove "fortunately"*

Done.

*L200: Did you validate your interpolation using the borehole measurements? If not, this could be a useful piece of analysis to validate your bedrock map and provide an indication of bedrock DEM uncertainty.*

Interesting idea, but two point measurement would probably not provide a statistically significant test on the bed rock interpolation or reveal reliable uncertainty estimate for it. We therefore prefer using it as part of the input data for the bedrock interpolation.

*L201-202: "has been constructed using the kriging interpolation method (processed using Surfer 13 © Golden Software LLC)"*

Done.

*L203-205: Combine "The filtered…at that time" with paragraph below.*

Done.

*L204-206: I would remove "An independent…(see section 4.1.3)" as it is not relevant here.*

Done.

*L216-217: "…were then differenced from the interpolated bedrock DEM to obtain…"*

Done.

*L217-219: "The lake outlines are converted to points and are prescribed a lake thickness of zero before interpolating each lake thickness map…"*

Replacement accepted with minor changes.

*L227: Change subheading to "Elevation changes and released volume of water during jökulhlaups in 2015 and 2018"*

Done.

*L229: Suggested change to "acquired by the TanDEM-x and TerraSAR-X spaceborne bistatic interferometer"*

This text was replaced with more simple but still accurate text.

*L231: "Differencing the two DEMs reveals…"*

Done.

*L235-236: Remove "a correction was deployed"*

Done.

*L236: "a ~500"*

Done.

*L238-239: "reference area and then subtracted from the elevation differences between the two DEMs."*

Sentences joint in slightly different manner.

*L241: "4 June during"*

Done.

*L259-264: Does the lake margin correspond to the area of elevation change in 2015? If so, you could use this to constrain the elevation change area in 2018.*

No it does not and does therefore not help constraining the elevation change area in 2018. We also prefer keeping those data sets (lowering data and RES) as independent as possible.

*L265-266: Where do these biases come from?*

Short explanation added. It should be admitted that this approximation is very crude. The main uncertainty in the lake volume drainage is however uncertainty of the crevasse volume estimate, particularly in 2015.

*L267269: Please state exactly why this correction needs to be applied.*

This text has been rephrased for clarification.

*L259-275: This feels like it should be in the results section. I suggest putting into the results under the heading "Water volume released during jökulhlaup in 2015 and 2018".*

The main purpose of the current text is explaining why we need to consider both types of volumes (even though they are given) and the uncertainty of these values. The volumes for the lake release are now first given in the results section (they were already were given there as well, hence no need to add specific section on that).

**Results**

*L279: "At the time of this observed maximum lake area in 2015,…"*

Done.

*L280-281: "In comparison, the lake had expanded to 3.2 km$^2$ in June 2018, two months priors to the 2018 jökulhlaup."*

Done.

*L281-283: Suggested change to: "The strong positive linear correlation between the area and volume of the subglacial lake is demonstrated in Fig. 6i.". This may also be impacted by the fact the same bedrock DEM is used throughout. Could the bedrock have changed over the acquisition period from e.g. subglacial volcanic activity?*

Text replaced according to suggestion. We respond to comments on assumption of fixed DEM in the general comments.

*L285: "lake volumes"*

Done.

*L290-291: Remove "indicating the applicability of our RES survey approach to evaluate the expected*

*hazard from a jokulhaup."*

Done.

*L292-314: This paragraph is interesting. You explain the offset of the 2014 and 2015 lake volumes extracted from the RES surveys by the different measurement densities, but 2016 and 2017 were also coarsely sampled. Could it be then that the offset is simply due to slightly different subglacial lake refilling rates? A better comparison would therefore be the linearly regress 2015-2017 and 2018-2020 separately. You could even try the same regression with the 2014-2015 data to see the result of this as well.*

These are good points. We however think it would not be very conclusive to split this as suggested particularly if only the RES-data are considered.  One reason is that the filling rate may differ, as you correctly point out, between different jökulhlaup cycle periods (2010-2015, 2016-2018 and 2019-2020), which do not coincide with the suggested periods. The main aim however with presenting these numbers is to get the average filling rate for our study period. We calculate this also without 2014 and 2015 RES-data point because they deviate slightly from the linear trend particularly when compared to the lake drainage volume in 2015 and we suggest an explanation why this could be. Even though the 2016 and 2017 were also obtained from more sparse profile grid we do not see a clear reason why the lake volume, then distributed in completely different way than in 2014 and 2015 should underestimated in a similar way. The particularly under-sampled area in 2014 and 2015, as pointed out later in this manuscript, is the northern part of the lake at these epochs, which the lake did not extend to in 2016 and 2017.

*L321: "The shape of the subglacial lake margin also differed between 2015 and 2018. "*

Done.

*L321-323: "Steep side walls surrounded the bulk of the lake, although the thickness of the lake was typically 10-30 m away from the lake margins (see Fig. 6f and left side of Fig. 7b)."*

The suggested text is not completely in accordance with what we are trying to say, hence this text has been rephrased for further clarification.

*L324-325: You should also reference Fig. 8a-b as this is where we see the greater lake thickness in 2018 – Fig.9a only shows the volume calculations.*

Reference added at an appropriate location in the sentence.

*L328-329: "The outward migration of the lake margin, typically by 50-150 m, was characterised by the outward propagation of the steep sided ice walls that defined the lake margin.".*

Replaced with shorter text mostly according to suggestion.

*L328-332: I would caution this discussion of the steep-sided subglacial lake walls. The interpolation was guided by setting the rim of the lake walls to zero. Whilst there is clear good evidence for steep-sided ice walls, there may also be interpolation error that could bias some of the lake shapes.*

We consider the points we make regarding the steep-sided lake not too strong, given the data we present. At locations in RES-profiles where the lake margin is pin-pointed we start getting reflection from bedrock hence lake thickness is zero and many of these profiles were surveyed close to perpendicular to the ice walls, hence side reflections (discussed in the manuscript later regarding shortcomings of 2D migration) should not skew significantly the location of lake roof reflections from the head of the walls. We therefore consider the evidences for these steep ice walls strong. Regarding the kriging interpolation, it tends to reduce the slope of these walls in between RES-profiles and can therefore be considered as potential source of underestimate of the volume, particularly when the distance is great between profiles (e.g. the northern part of the lake in 2014 and 2015). Regarding this particular sentence, commented on here, the part of it which dealt with the ice walls has been omitted since it was partly repeating previously mentioned observations.

*L353-355: It appears to me that the maximum surface lowering regions are at the centre where the lake is not at its thickest – the largest lake thickness is to the eastern side of the ESC.*

A thorough discussion on why the location of thickest water found at the eastern part of the cauldrons is not directly mirrored as the maximum lowering is found in section 4.3.

*L355-369: This paragraph is a useful summary, but I think you the need make it clear what the terms 'thickening' and 'thinning' are referring to. If, as is stated in text, this refers to changes in ice thickness, you should also then state that this assumes the lake has completely drained. This means that for a completely drained lake, if the difference between the lake thickness and the surface lowering is positive, then the ice has thickened. I would try to reword this paragraph to make this more clear. Do you have additional data to suggest that the entire lake has drained?*

That's a good point. Section 3.1 now includes a justification why we think the lake drained completely or was reduced to an insignificant volume. We also added a short text to this paragraph explaining that the lake "leftovers" should therefore not be a problem for our ice thickening/thinning estimate.

**Discussion**

*L371-383: This paragraph provides results that are then discussed in the subsequent sections. To improve the flow of the text, it might better to have this as a separate section (4.1.5.) and use it to summarise the contributing errors and state the accuracy of 10-20%. The authors may leave this suggestion if they do not see it as useful.*

To clarify further that the uncertainties in our RES deduced lake volumes are obtained by comparison with independent estimates of lake volume drainage during jökulhlaups, the main content of this paragraph has been moved to more appropriate sections (method section 2.5 and result section 3.1). There the content of this paragraph has been further extended including further justifications on why we consider this validation is applicable. The remaining part of the paragraph is introduction to the content of section 4.1.

*L374-379: This is a long sentence. Suggest shortening: "The RES surveys before the jökulhlaups in 2015 and 2018 show a good agreement with the derived surface lowering patterns (Fig. 8–9). Together with a close linear relationship between the time elapsed since the previous jökulhlaup (Fig. 9b) except when the lake is small and not hazardous suggests lake volume errors from RES measurements are typically 10-20%."*

The content of this text has been rephrased and moved to section 3.1.

*L394: Has the 20% error been calculated based on the data gaps created from the supraglacial lakes? You should state this here to be clear.*

No, and we hope this is clear from the current text.

*L393-399: Even though this discussion refers to RES data gaps, I think it is best placed under Section 4.1.4 and I would suggest moving it there.*

We have now taken what used be section 4.1.4 and merged most of it with section 4.1.3 (the part which dealt with temporal variation in $c_{gl}$) and the rest with section 4.2 (regarding subglacial and englacial water bodies as potential jökulhlaup trigger). Since there is no longer a specific section subglacial and englacial water bodies we consider this now as the appropriate location of this text.

*L401-404: Suggested shortening to: "In most glaciological applications, only 2D migration of RES data is possible but even this requires the assumption that all radar reflections originate from directly beneath the survey profile. This is often not the case beneath glaciers that flow over volcano's, where the subglacial topography is particularly complex."*

Suggested text used with minor changes.

*L405: Remove "but much smaller when the slope and profile directions are in parallel" and combine with sentence spanning Lines 405-406.*

Done.

*L406-410: "If the traced reflective surface is not directly beneath the RES profile but cross-track, the obtained ice thickness is underestimated and the mapped surface below the profile is estimated to be too high."*

Done.

*L410: "This has been shown using an experiment…"*

Done.

*L411-412: "Similar results were obtained in a recent study on Mýrdalsjökll ice cap (S-Iceland) which has a similar topographic setting to the ESC (Magnússon et al., in review). In that study, 2D migrated profiles were found to be 10 m higher than the bedrock DEM obtained from 3D migrated data."*

The suggested text was used with minor changes. We also deleted the next 4 lines following this which included unnecessary details on the referenced study.

*L418: "allow for safe acquisition of data for 3D migration without reasonable effort"*

This suggested minor change overturns the meaning of the sentence, compared to the intended one. We, however, do not see what is wrong with the original text and therefore it is unchanged.

*L419-421. The two effects probably mostly cancel each other out, but given the complexity of the subglacial topography, I'm not sure this can be stated with confidence. I would instead note that in general the two effects cancel out, but across the complex topography described in this study, the effect is likely to be similar to the Magnússon et al. (in review) study.*

This is now stated with less confidence ("..this may to a large extent be cancelled out.." replaced for "..this may to a some extent be cancelled out.." ) and we also point out that magnitude of the errors caused by the shortcoming of the 2D migration is likely to vary between surveys for the lake roof.

*L423-428: I think this paragraph should be placed in Section 4.1.1. as it discusses the effects of data gaps on underestimating subglacial topographic peaks. Lines 424-425 can be retained in this section as it is pertinent to the effects of the 2D migration processing.*

We think the current location of this text is more appropriate even though we agree it could fit in the other section as well. Due to the nature of the survey we are probably receiving reflections from many of the peaks in the lake roof, even though we are not driving directly above them. Due to the steep topography of the lake roof these peaks probably often correspond to the nearest reflective surface of the lake roof, particularly after the profile separation was reduced to 200-250 m. The locations of peaks in the lake roof derived from the 2D migrated RES profiles are therefore not just prone to under-sampling of profiles, for this steep topography the limitation of the 2D migration are probably even more important.

*L432: Remove "the"*

Done, assuming you are referring to the "the" in front of account (there were two more "the" to choose from).

*L433: "that the actual"*

Done.

*L440-441: "shift both the lake roof and the bedrock in the same…"*

Done.

*L443-446: I think this section needs rewording. Suggestion: "If $c_{gl}$ is too large, lake thickness is overestimated, and it is underestimated if $c_{gl}$ is too low. For example, if the true value of $c_{gl}$ is $1.60 \times 10^8$ m s$^{-1}$ but a value of $1.68 \times 10^8$ m s$^{-1}$ is used, the lake volume would be overestimated by ~5%. Considering the upper limit of $c_{gl}$ is $1.70 \times 10^8$ m s$^{-1}$, a significant overestimate of $c_{gl}$ is unlikely."*

Rephrased in a similar way as suggested. We apologise for using incorrectly "overestimate" instead of underestimate at one location in the old text here, which likely caused confusion.

*L451: "…value of $c_{gl}$ may differ significantly between some lake roof and bedrock measurements, leading to larger uncertainties at locations where such water bodies exist."*

Reworded in similar manner as suggested.

*L452-454: Are crevasses persistent around the cauldron rim or are they transient features? Are they seasonally filled in with snow?*

Both, the largest ones leave open scars in the glacier surface (probable only to shallow depth), which even survive between jökulhlaups, while the smaller ones close soon after each jökulhlaup (both by snow and compression). The large persistent crevasses at the rim are far outside the area where the supraglacial lake is formed.

*L468: "hilly topography" is a little colloquial, I suggest changing to "undulating lake surface topography"*

Done.

*L475: Remove "minority" to "small number".*

Replaced for "small part" (part is better since part of individual profiles could be traced).

*L485-492: This is a worthwhile discussion but should be moved, possibly to Section 4.3.*

We moved this to end of section 4.2.

*L500-504: I don't think this part is necessary, I would focus on the results you have presented in the paper and use these to develop your discussion.*

Text omitted.

*L505-507: Reword to: "Temperature profiles within the subglacial lakes beneath the Skaftá cauldrons have revealed temperatures of 3-5°C that are mostly independent of lake depth, thus enabling effective convection to take place (Jóhannesson et al., 2007; unpublished data at the IMO). Chemical…".*

Done.

*L509: Remove "a".*

Done.

*L518-519: Here, you suggest enhanced geothermal activity is likely the reason for enhanced basal melting. Do you have additional data/publications to back this up? Could other factors play a role?*

It was maybe not clear here that this sentence was directly connected to the previous one. By starting the sentence with "Such temporal increase.." should have clarified this. We are here discussing what could cause the temporal changes in the shape of the lake from one jökulhlaup period to another and to us this is the most likely factor. Below we also point out that the longer developing time in 2010-2015 may also play role even though we doubt that this alone can explain observed difference in lake shape between 2015 and 2018. We prefer not extending this already long discussion with text ruling out far less likely explanations. Regarding possible volcanism, which you might be referring to here we refer to answer to a specific comment above.

*L558-359: "The GNSS station based on the ESC surface, operated by IMO, was operating during both jökulhlaups."*

Since you ask about the position of the station below regarding Figure 10, we think it is actually better to include the statement here that they were at approximately the same position. We however omitted the word "on-line".

*L539-541: I suggest remove this sentence as it is redundant.*

Done.

*L547-549: "The differences in lake area between 2015 and 2018 can explain the difference in surface uplift rates but not the slower initial subsidence in 2015".*

Done including minor additions.

*L550-561: This paragraph could be combined with the preceding paragraph and shortened.*

The paragraphs combined and slightly shortened.

*L564-568: "Such a deceleration is not observed in 2015 and may be caused by floating ice atop the 10-20 m thick water layer moving against the bedrock a few hundred metres south of the GNSS station. Whilst a supraglacial lake inhibited complete mapping south of the GNSS station, traced reflections from RES data 450 m south of the station suggest the ice was grounded at this location.".*

Done including minor additions.

*L574: I'm not sure I see where the 2018 subsidence sped up again. Is it just before the 4-day mark? Either way, it seems relatively insignificant – the fact it is not apparent in the 2018 data should be highlighted as a difference between 2015 and 2018.*

It is small but significant considering the accuracy of the subsidence. The scale on the y-axis is however not favourable here, but kept as it is to better show other events. We do however highlight how small this sped up was in 2018.

*L585: Remove "near"*

Done.

*L593: "cauldron centre to be substantially…"*

Done.

*L595: "of ice at a given"*

Done.

*L596-598: The thickening is also partially due to the convergence of ice flow into the cauldron and should acknowledged here.*

This is highlighted in next paragraph.

*L607: "motion decreases from"*

*L616-631: I think this is an interesting place to end, but I would also add that the RES survey design could also be improved so that the subglacial lake can be mapped at sufficient resolution to remove interpolation errors.*

There is a limit on how more profiling would improve our results without applying 3D migration (which is simple too much effort and such data set is often impossible to obtain due to crevasses as mention in section 4.1.2). We will at certain point reach a limit where the governing errors are caused by the shortcoming of the 2D migrations rather than the interpolation. In the comparison of 2D and 3D migrated results from Mýrdalsjökull (Magnússon et al., in press) referenced in section 4.1.2 the interpolated DEM obtained from the 2D migrated profile had similar errors when compared to the DEM from 3D migrated data, both in terms of mean bias and standard deviation of the errors in the traced bedrock elevation of the 2D migrated profiles at the profile locations. This suggest that in this case the errors due to the shortcoming of the 2D migration where the governing error not the interpolation. For what profile separation this limit is reached, surely depends on local topographic setting as well as on how well the profiles are located in order to minimise the 2D migration error, but in this particular study (Magnússon et al., in press) the separation between the 2D migrated profiles was 200 m. We now shortly mention this in section 4.1.2.

**Conclusions**

*L633-634: "Repeat RES surveys over the Eastern Skaftá Cauldron (ESC) and a comparison with surface lowering during jökulhlaups was used to measure the volume of a subglacial lake beneath the ESC every year between 2014 and 2020. This novel data set has been used to demonstrate the applicability of repeat RES surveys for quantitative monitoring of subglacial lake volumes."*

We try to highlight the validation as the reason for our statement on the applicability of the RES for this monitoring, hence the current text resembles more the old one than the one suggested hear above.

*L632-646: This is more of a summary than a conclusion. I think some key information is missing. For example, what are the uncertainties of the RES data, by how much has the lake volume changed over time and information relating to the two jökulhlaups (2015 and 2018). You should then frame these key results into a brief summary of the advantages and limitations of the repeat-RES approach and suggest possible future developments of the technique.*

Given that the paper is already quite long, we rather prefer not repeating in the conclusions values both given in results and abstract. We however added lines regarding why we think it is hard improve this method for ESC but state also where improved methodology adopting 3D migration of the RES data can be applied.

**References**

*L766: Change "USAGE" to "ISAGE"*

Done.

**Figures**

*Figure 1: Good figure overall. Could you label the red boxes in panel (a) to make it clear which each is referring to (i.e. red box on inset panel is referring to panel (a) and inset in panel (a) is referring to panels (b) and (c))? Could you label ESC and WSC instead of "Skaftá cauldrons"? Is the glacier outline from GLIMS and does it need a reference? For panels (d) and (e), it is best to have the dates on the images to avoid excessive reference to the figure caption, with exact dates if you have them. For demonstrating the viewing angle, I think you should mark at the apex of the red line that this is the position of the camera, with an arrow indicating viewing direction.*

The inset is now labelled with b and the references of the red boxes in a and b (formerly referred to as inlet) clarified. The labels ESC and WSC have been replaced for *"Skaftá cauldrons"* and the dates of the photograph added. We decided not to replace the dashed red lines with apex and arrow lines since the apex of photograph e is outside the span of image c. The outlines are home-maid, digitised by some of the authors from various sources, representing Vatnajökull around 2000. This is however of little importance for the study, hence we do not include any details on this.

*Figure 2: Interesting figure. The red box on the inset map of Vatnajökull is annotated "ESK" which I assume should be "ESC". In panel (a), I worry that red and green lines cannot always been seen by those with colour blindness, possibly change to blue or black? I would change the map of Vatnajökull to panel (a) and have the map of the profile grid as panel (b). Thus, change "(located on corner inlet)" to "(location in **a**). The legend details are a bit scattered. It might be better to put all of these (e.g. bed reflection, surfaces) at the bottom of each panel to avoid cluttering the graphs.*

Changed according to all suggestions.

*Figure 3: Good figure*

*Figure 4: Overall a useful figure. I wonder if panel **h** would be more instructive if it showed the survey lines in the same colour e.g. black, or possibly showing the sparse (2014-2018) and dense (2018-2020) survey lines in different colours.*

We tried this figure using only single black colour on **h**. This sort of made the different colour shown for the other years pointless (would also be so with two colour approach). We however think that by using the multicolour approach, we better indicate that the profiles are quite accurately repeated from one year to another, hence we prefer sticking with the old version here.

*Figure 5: Good figure*

*Figure 6: Good figure overall. Scale bar needs a label "Lake Thickness (m)" and the bedrock DEM also needs a corresponding color scale. In the figure caption, change "ESK" to "ESC". Red might not be the best colour to use for the survey lines, suggest change to dotted black (or another suitable colour).*

Bedrock colour scale and labels added and text replaced. Black survey lines tends to merge with black contours line (showing both bedrock elevation and lake thickness) and therefore we stick with the red survey lines.

*Figure 7: Very interesting figure. My only concern is that the figure suggests the surface lowering occurs without a change in the lake water level. You could caution this on the figure by writing the date the lake volume has been estimated.*

Annotation added to the figure to clarify this.

*Figure 8: Each color scale should have a label. The red and cyan lines could also be annotated on the map rather than having to refer to the figure caption. Otherwise, a good overview of the surface changes observed.*

Changed according to suggestion. Further explanation on the dashed red line is however kept in caption.

*Figure 9: I have no major problems with this figure. If possible, it would be best to move the legend above the figure to avoid overfilling panel (a).*

We tested that but we did not think it improve the appearance of the Figure, hence it is unchanged.

*Figure 10: Was the GNSS repositioned to the exact same location before each jökulhlaup event? I would also state what the symbols mean in the figure caption or legend to make it easier for the reader to understand.*

As stated in the main text the station was at approximately the same location during both jökulhlaups. It was re-deployed once during the 3 year period but at approximately same position as it used to be. I assume that you are asking because you are interested in the ice motion in between jökulhlaups. Even though this motion is interesting it is of little relevance for this study, hence we do not include this info. We have clarified further that the forms indicate timestamps of events discussed in section 4.3. Adding caption or legend saying what these events would require a far too long text to fit in a legend or substantially extend the figure caption in a manuscript, which already quite long.

---

## Author Comment (AC2)

**Response to review by Anonymous Referee #1 of manuscript TC-2021-65:**

**Development of a subglacial lake monitored with radio echo sounding: Case study from the Eastern Skaftá Cauldron in the Vatnajökull ice cap, Iceland**

*This is a nicely presented and well written paper, covering a topic suitable for publication in The Cryosphere. The approach is valid and clearly explained; conclusions are supported by results; and study limitations are considered in appropriate detail. I suggest publication subject to the very minor edits listed below.*

*Ln 44. 'a depression' is singular, 'ice cauldrons' is plural. I suggest 'an ice cauldron'.*

Sentence has been changed according to suggestion by reviewer 1.

*Ln 52. Replace 'date at back' with 'date back'.*

Done.

*Line 77. Replace 'made it possible to map of the' with 'made it possible to map the'.*

Sentence has been changed according to suggestion by reviewer 1.

*Line 78. Delete 'possible' (following the word 'observations').*

Sentence has been changed according to suggestion by reviewer 1.

*Line 84. Replace 'not be a clear' with 'not a clear'.*

Sentence has been changed according to suggestion by reviewer 1.

*Line 102. Replace 'as well as an annual estimate the area' with 'as well as annual estimates of the area'.*

Sentence has been changed according to suggestion by reviewer 1.

*Line 110. Replace 'Eastern Skaftá Cauldron' with ESC.*

Done.

*Line 116. Replace 'were defect' with 'were defective' [or something similar].*

Changed according to suggestion. This text has however, now been moved to a table according to suggestion from reviewer 1.

*Line 124. Replaces 'by towing with a snowmobile a low frequency transmitter' with 'by towing a low frequency transmitter with a snowmobile'.*

Sentence has been changed according to suggestion by reviewer 1.

*Line 128-130. This sentence (starting 'The y-axis'…) is unclear. I suggest re-wording.*

This text has been re-written.

*Line 134. Delete 'the' (before the word 'half').*

Done.

*Line195. Delete 'now' (before the word 'split').*

Done.

*Line 197. Delete 'now' (before the word 'merged').*

Done.

*Line 216. Delete 'now' (before the word 'compared').*

This text has been rewritten.

*Line 217. The word lake 'thickness' is used here and throughout the manuscript. Though a subglacial lake, is 'depth' more appropriate than 'thickness'?*

We agree that the word "thickness" is strange. However the word "depth" sort of indicates something relative to a fixed flat upper surface (as for a normal lake), which is not at all the case for the roof of the subglacial lake beneath ESC. For this reason we find the word "depth" misleading and therefore worse than the term "thickness" and we have not found a good and more appropriate replacement.

*Line 277. Replace 'areas of' with 'area of'.*

Done.

*Line 287. Replace 'August same' with 'August the same'.*

Done.

*Line 294. Replace 'possible' with 'possibly'.*

Done.

*Line 296. Replace 'volume few' with 'volume a few'.*

Done.

*Line 321-21. Replace 'east of boreholes' with 'east of the boreholes'.*

Done.

*Line 354. Replace 'approximately same' with 'approximately the same'.*

Done.

*Line 390. 'observations on roof elevation there is also an observation on bedrock' with 'observations of roof elevation there is also an observation of bedrock'.*

Done.

*Line 584-585. Delete the second 'near' in the sentence: 'The ice surface geometry near the station near in the village'.*

Done.

*Line 634. Replace 'on the lake volume' with 'of the lake volume'. Also, this is the first point in the conclusions where 'the lake' is mentioned. It might be worth stating at this point that you are referring to a subglacial lake beneath an ice cauldron (rather than saving this to later in the conclusions).*

Done. The text has also been rephrased to mention the subglacial lake beneath the first cauldron in the first sentences.

*Line 365. Replace 'a key knowledge' with 'key knowledge'.*

Done.

*Figure 1 caption. Replace 'inlet' with 'inset'.*

Inset now labelled with 'b'. The word 'inlet' therefore omitted.

*Figure 2 caption. Replace 'inlet' with 'inset'.*

Inset now labelled with 'b'. The word 'inlet' therefore omitted.

*Figure 6 caption. Replace 'ESK' with 'ESC'.*

Done.

---

## Author Response (AR2)

**Response letter**

Thanks to the reviewer for these useful comments shown with italic font below, followed with explanation (not with italic font) on how we responded to them. Also thanks to the editor for his comments, which were also very useful but the author responses to his comments and are incorporated as replies to editors comments in the file tc-2021-65-comments-to-author-1.pdf further below. At the end the new version of the manuscript is attached showing the changes from the manuscript submitted in June.

*L29: "a lake has previously caused"*

I am not sure what "previously" should refer to here, this was not observed or linked to a jökulhlaup happening afterwards (actually the leakage happened after a large jökulhlaup, damaging the ceil of the lake). I therefore think the reviewer may be misunderstanding the sentence and it has therefore been rephrased.

*L48: Suggest minor rephrasing: "Grímsvötn has been known to exist as a lake within Vatnajökull for centuries and has previously been the source of large jökulhlaups draining from the Skeiðarárjökull outlet glacier in S-Vatnajökull, although its exact location was not well known until identified in an expedition in 1919 (Wadell, 1920)."*

Rephrased similarly as suggested but with shorter text.

*L121: "strength of both the…"*

Done.

*L123: "(using cair=3.0x108 m s-1), which yields the" (this section is much clearer now, thank-you for amending this).*

Done.

*L241: "was more than a few decimetres…"*

Done.

*L278: "which are difficult to quantify, can…"*

Done. The word "difficult" in next line was therefore replaced with "problematic".

*L294: "prior to each jökulhlaup"*

Done

*L333: "large proportions"*

Done.

*L343: "extended a few hundred"*

Done.

*L367: "This further suggests that the"*

Done.

*L479: "obstruct reflections from"*

Done.

*L638: "in 2014-2020 for quantitative"*

Done.

*L639: "cauldron, validated with calculated surface lowering"*

Done except the word "calculated" was replaced with "observed".

[revised manuscript text omitted]